# UNSSOR: Unsupervised Neural Speech Separation by Leveraging Over-determined Training Mixtures

**Zhong-Qiu Wang** and **Shinji Watanabe**
Language Technologies Institute, Carnegie Mellon University, Pittsburgh, USA
wang.zhongqiu41@gmail.com

## Abstract

In reverberant conditions with multiple concurrent speakers, each microphone acquires a mixture signal of multiple speakers at a different location. In over-determined conditions where the microphones out-number speakers, we can narrow down the solutions to speaker images and realize unsupervised speech separation by leveraging each mixture signal as a constraint (i.e., the estimated speaker images at a microphone should add up to the mixture). Equipped with this insight, we propose UNSSOR, an algorithm for unsupervised neural speech separation by leveraging over-determined training mixtures. At each training step, we feed an input mixture to a deep neural network (DNN) to produce an intermediate estimate for each speaker, linearly filter the estimates, and optimize a loss so that, at each microphone, the filtered estimates of all the speakers can add up to the mixture to satisfy the above constraint. We show that this loss can promote unsupervised separation of speakers. The linear filters are computed in each sub-band based on the mixture and DNN estimates through the forward convolutive prediction (FCP) algorithm. To address the frequency permutation problem incurred by using sub-band FCP, a loss term based on minimizing intra-source magnitude scattering is proposed. Although UNSSOR requires over-determined training mixtures, we can train DNNs to achieve under-determined separation (e.g., unsupervised monaural speech separation). Evaluation results on two-speaker separation in reverberant conditions show the effectiveness and potential of UNSSOR.

## 1 Introduction

In many machine learning and artificial intelligence applications, sensors, while recording, usually capture a mixture of desired and undesired signals. One example is the cocktail party problem (or speech separation) [1, 2], where, given a recorded mixture of the concurrent speech by multiple speakers, the task is to separate the mixture to individual speaker signals. Speech separation [3] has been dramatically advanced by deep learning, since deep clustering [4] and permutation invariant training (PIT) [5] solved the label permutation problem. They (and their subsequent studies [6–26]) are based on supervised learning, requiring paired clean speech and its corrupted signal generated via simulation, where clean speech is mixed with, for example, various noises and competing speakers at diverse energy and reverberation levels in simulated rooms [3]. The clean speech can provide an accurate, sample-level supervision for model training. Such simulated data, however, may not match the distribution of real-recorded test data in the target domain, and the resulting supervised learning based models would have generalization issues [27, 28]. How to train unsupervised neural speech separation systems on unlabelled target-domain mixtures is hence an important problem to study.

Training unsupervised speech separation models directly on monaural mixtures is an ill-posed task [2], since there is only one mixture signal observed but multiple speaker signals to reconstruct. The separation model would lack an accurate *supervision* (or regularizer) to figure out what desired sound objects (e.g., clean speaker signals) are, as there are infinite solutions where in each solution the

37th Conference on Neural Information Processing Systems (NeurIPS 2023).

estimated sources can sum up to the mixture. Supposing that the separation model does not separate well and outputs a clean speaker signal plus some competing speech, noise or reverberation, would this output be viewed as a desired sound object? This is clear to humans, clear to supervised learning based models (by comparing the outputs with training labels), but not really clear to an unsupervised model. On the other hand, many studies [3–26] have observed that deep learning based supervised learning can achieve remarkable separation. In other words, with proper supervision, modern DNNs are capable of separating mixed speakers, but, in an unsupervised setup, there lacks an accurate supervision to unleash this capability. The key to successful unsupervised neural separation, we believe, is designing a clever *supervision* that can inform the model what desired sound objects are, and penalize the model if its outputs are not good and reward otherwise.

Our insight is that, in multi-microphone over-determined conditions where the microphones outnumber speakers, the ill-posed problem can be turned into a well-posed one, where a unique solution to the speakers exists (up to speaker permutation). This well-posed property (that a unique solution exists) can be leveraged as a *supervison* (or regularizer) to design loss functions that could inform the unsupervised separation model what desired sound objects are and promote separation of speakers.

Equipped with this insight, we perform unsupervised neural speech separation by leveraging multi-microphone over-determined training mixtures. Our DNNs can be trained directly on over-determined mixtures to realize over- and under-determined separation. The proposed algorithm, named UNSSOR, obtains strong separation performance on two-speaker separation. Our contributions include:

- We enforce a linear-filter constraint between each speaker's reverberant images at each microphone pair, turning the ill-posed problem into a well-posed one that can promote separation of speakers.
- We formulate unsupervised neural speech separation as a blind deconvolution problem, where both the speaker images and linear filters need to be estimated. We design loss functions motivated by the blind deconvolution problem, and propose a DNN approach to optimize the loss functions, where the speaker images are estimated via DNNs and the linear filters are estimated via a sub-band linear prediction algorithm named FCP [29] based on the mixture and DNN estimates.
- We propose a loss term, which minimizes a metric named intra-source magnitude scattering, to address the frequency permutation problem incurred when using sub-band FCP.
- Based on over-determined training mixtures, UNSSOR can be trained to perform under-determined separation (e.g., monaural unsupervised speech separation).

## 2 Related work

Various unsupervised neural separation algorithms, which do not require labelled mixtures, have been proposed. The most notable one is mixture invariant training (MixIT) [30–34], which first synthesizes training mixtures, each by mixing two existing mixtures, and then trains a DNN to separate the resulting mixed mixtures to underlying sources such that the separated sources can be partitioned into two groups and the separated sources in each group can sum up to one of the two existing mixtures (used for mixing). Care needs to be taken when synthesizing mixtures of mixtures. First, the sources in an existing mixture could have similar characteristics (e.g., similar reverberation patterns as the sources in an existing mixture are recorded in the same room) that are informative about which sources belong to the same existing mixture, and this would prevent MixIT from separating the sources [30, 35]. Second, it is unclear how to mix existing multi-channel mixtures, which are usually recorded by devices with different microphone geometry and number of microphones. Third, mixing existing mixtures with different reverberation characteristics would create unrealistic mixtures.

UNSSOR avoids the above issues by training unsupervised neural separation models directly on existing mixtures rather than on synthesized mixtures of mixtures. An earlier study related to this direction is the reverberation as supervsion (RAS) algorithm [36], which addresses monaural two-speaker separation given binaural (two-channel) training mixtures. RAS performs magnitude-domain monaural separation directly on the left-ear mixture and then linearly filters the estimates through time-domain Wiener filtering so that the filtered estimates can approximate the right-ear mixture. RAS essentially does monaural separation and is effective at separating speakers in a semi-supervised learning setup, where a supervised PIT-based model is first trained and then used to bootstrap unsupervised training. It however fails completely in fully-unsupervised setup [36], unlike UNSSOR.

Conventional algorithms such as independent component analysis [37–41], independent vector analysis (IVA) [41–44] and spatial clustering [45–48] can perform unsupervised separation directly on existing mixtures. They perform separation based on a single test mixture at hand and are not

designed to learn speech patterns from large training data, while UNSSOR leverages DNNs to model speech patterns through unsupervised learning, which could result in better separation. Another difference is that UNSSOR can be configured for monaural, under-determined separation, while ICA, IVA and spatial clustering cannot. There are studies [49–52] training DNNs to approximate pseudo-labels produced by conventional signal processing based separation models such as spatial clustering and blind source separation (BSS) algorithms. Their performance is however often limited since spatial clustering and BSS themselves are not good enough at separation. There are studies [53–55] training DNNs to separate multi-speaker mixtures such that the likelihood of observed mixtures under a probabilistic distribution derived based on the DNN separation results can be maximized. Such methods rely on statistical models for separation. They require costly iterative estimation of signal statistics at run time. In addition, the DNNs are only leveraged in estimating target magnitude, while phase estimation is only realized by spatial filtering.

## 3  Problem formulation

Given a $P$-microphone mixture with $C$ speakers in reverberation conditions, the physical model can be formulated using a system of linear equations in the short-time Fourier transform (STFT) domain:

$$Y_p(t, f) = \sum_{c=1}^{C} X_p(c, t, f) + \varepsilon_p(t, f), \text{ for } p \in \{1, \ldots, P\}, \tag{1}$$

where $t$ indexes $T$ frames, $f$ indexes $F$ frames, and at microphone $p$, time $t$ and frequency $f$, $Y_p(t, f)$, $X_p(c, t, f)$ and $\varepsilon_p(t, f) \in \mathbb{C}$ respectively denote the STFT coefficients of the mixture, reverberant image of speaker $c$, and non-speech signals. In the rest of this paper, we refer to the corresponding spectrogram when dropping the index $c$, $p$, $t$ or $f$. We assume that $\varepsilon$ is weak and stationary (e.g., a time-invariant Gaussian noise or simply modelling errors). Without loss of generality, we designate microphone 1 as the reference microphone. Our goal is to, in an unsupervised way, estimate each speaker's image at the reference microphone (i.e., $X_1(c)$ for each speaker $c$) given the input mixture. We do not aim at dereverberation, instead targeting at maintaining the reverberation of each speaker.

Unsupervised separation based only on the observed mixture is difficult. There are infinite solutions to the above linear system where there are $T \times F \times P$ equations (we have a mixture observation for each $Y_p(t, f)$) but $T \times F \times P \times C$ unknowns (we have one unknown for each $X_p(c, t, f)$).

Our insight is that the number of unknowns can be dramatically reduced, if we enforce constraints to the speaker images at different microphones. Since $X_1(c)$ and $X_p(c)$ are both convolved versions of the dry signal of speaker $c$, there exists a linear filter between them such that convolving $X_1(c)$ with the filter would reproduce $X_p(c)$. This convolutive relationship is a physical constraint, which can be leveraged to reduce the number of unknowns. Specifically, we formulate (1) as

$$Y_1(t, f) = \sum_{c=1}^{C} X_1(c, t, f) + \varepsilon_1(t, f),$$
$$Y_p(t, f) = \sum_{c=1}^{C} \mathbf{g}_p(c, f)^{\mathsf{H}} \, \widetilde{\mathbf{X}}_1(c, t, f) + \varepsilon_p'(t, f), \text{ for } p \in \{2, \ldots, P\}, \tag{2}$$

where $\widetilde{\mathbf{X}}_1(c, t, f) = [X_1(c, t - A, f), \ldots, X_1(c, t, f), \ldots, X_1(c, t + B, f)]^{\mathsf{T}} \in \mathbb{C}^{A+1+B}$ stacks a window of $E = A + 1 + B$ T-F units, $\mathbf{g}_p(c, f) \in \mathbb{C}^E$ is the *relative room impulse response* (relative RIR) relating $X_1(c)$ to $X_p(c)$, and $(\cdot)^{\mathsf{H}}$ computes Hermitian transpose. $\mathbf{g}_p(c, f)$ is not long (i.e., $E$ is small) [56] if microphone 1 and $p$ are placed close to each other, which is the case for compact arrays.

An implication of this constraint is that the number of unknowns is reduced from $T \times F \times P \times C$ to $T \times F \times C + F \times (P - 1) \times E \times C$[1], which can be smaller than the number of equations (i.e., $T \times F \times P$) when $P > C$ (i.e., over-determined conditions) and when $T$ is sufficiently large (i.e., the input mixture is reasonably long). In other words, this formulation suggests that (1) there exists a solution for separation, which is most consistent with the above linear system; and (2) in over-determined cases, it is possible to estimate the speaker images in an unsupervised way.

---

[1]$T \times F \times C$ is because there is one unknown for each $X_1(c, t, f)$, and $F \times (P - 1) \times E \times C$ is because $\mathbf{g}_p(c, f)$ is $E$-tap and we have one such filter for each of $P - 1$ microphone pairs for each frequency and speaker.

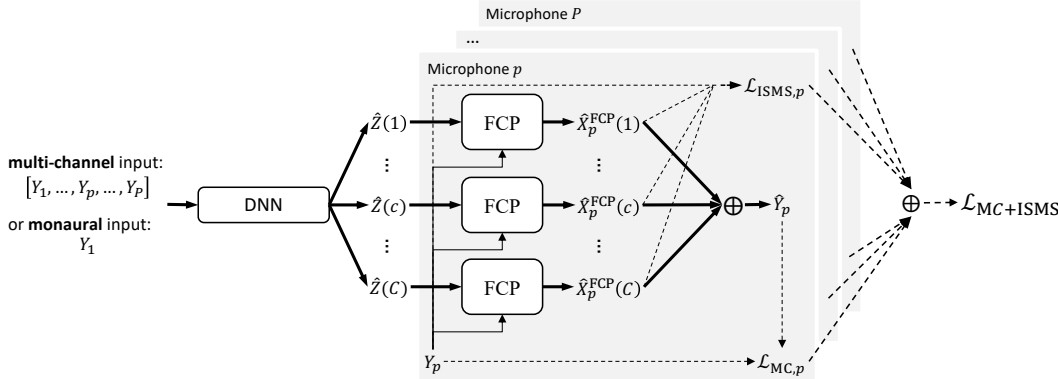

Figure 1: Illustration of UNSSOR (assuming $P > C$ during training).

As $\varepsilon$ is assumed weak, time-invariant and Gaussian, one way to find the solution is to compute an estimate that is most consistent with the linear system in (2) by solving the following problem:

$$\underset{\mathbf{g}.(\cdot,\cdot),X_1(\cdot,\cdot,\cdot)}{\operatorname{argmin}} \sum_{t,f} \left| Y_1(t,f) - \sum_{c=1}^{C} X_1(c,t,f) \right|^2 + \sum_{p=2}^{P} \sum_{t,f} \left| Y_p(t,f) - \sum_{c=1}^{C} \mathbf{g}_p(c,f)^{\mathsf{H}} \widetilde{\mathbf{X}}_1(c,t,f) \right|^2. \tag{3}$$

This is a blind deconvolution problem [57], which is non-convex in nature and difficult to be solved if no prior knowledge is assumed about the relative RIRs or the speaker images, because both of them are unknown. In the next section, we propose a DNN-based approach, which can model speech patterns through unsupervised learning (and hence model speech priors), to tackle this problem.

## 4 Method

Fig. 1 illustrates the proposed system. The DNN takes in the mixture at all the $P$ microphones or at the reference microphone 1 as input and produces an intermediate estimate $\hat{Z}(c)$ for each speaker $c$. FCP [29] is then performed on $\hat{Z}(c)$ at each microphone $p$ to compute a linear-filtering result, denoted as $\hat{X}_p^{\mathrm{FCP}}(c)$, which, we will describe, is essentially an estimate of the speaker image $X_p(c)$. After that, two loss functions are computed and combined for DNN training. This section describes the DNN configuration, loss functions, FCP filtering, and an extension for monaural separation.

### 4.1 DNN configurations

The intermediate estimate $\hat{Z}(c)$ for each speaker $c$ is obtained via complex spectral mapping [23, 58], where we stack the real and imaginary (RI) parts of the input mixture as features for the DNN to predict the RI parts of $\hat{Z}(c)$. For the DNN architecture, we employ TF-GridNet [23], which obtains strong results on supervised speech separation benchmarks. See Appendix I for more DNN details.

### 4.2 Mixture-constraint loss on filtered estimates

Following formulation in (3), we propose *mixture-constraint* (MC) loss, which is computed by filtering the DNN estimate $\hat{Z}(c)$ of each speaker $c$ to approximate the $P$-channel input mixture:

$$\mathcal{L}_{\mathrm{MC}} = \alpha_1 \sum_{t,f} \mathcal{F}(Y_1(t,f), \sum_{c=1}^{C} \hat{Z}(c,t,f)) + \sum_{p=2}^{P} \alpha_p \sum_{t,f} \mathcal{F}(Y_p(t,f), \sum_{c=1}^{C} \hat{\mathbf{g}}_p(c,f)^{\mathsf{H}} \widetilde{\hat{\mathbf{Z}}}(c,t,f)). \tag{4}$$

In (4), $\widetilde{\hat{\mathbf{Z}}}(c,t,f)$ stacks a window of T-F units around $\hat{Z}(c,t,f)$, and $\hat{\mathbf{g}}_p(c,f)$ is an estimated relative RIR computed based on $\hat{Z}(c,\cdot,f)$ and the mixture $Y_p(\cdot,f)$ through FCP [29]. Both of them will be described in the next sub-section. $\alpha_p \in \mathbb{R}$ is a weighting term for microphone $p$. Following [23], $\mathcal{F}(\cdot,\cdot)$ in (4) computes an absolute loss on the estimated RI components and their magnitude:

$$\mathcal{F}\left(Y_p(t,f), \hat{Y}_p(t,f)\right) = \frac{1}{\sum_{t',f'} |Y_p(t',f')|} \left( \left| \mathrm{Re}(Y_p(t,f)) - \mathrm{Re}(\hat{Y}_p(t,f)) \right| \right.$$
$$\left. + \left| \mathrm{Im}(Y_p(t,f)) - \mathrm{Im}(\hat{Y}_p(t,f)) \right| + \left| |Y_p(t,f)| - |\hat{Y}_p(t,f)| \right| \right), \tag{5}$$

where $\text{Re}(\cdot)$ and $\text{Im}(\cdot)$ respectively extract RI components and $|\cdot|$ computes magnitude. The term $1/\sum_{t',f'}|Y_p(t',f')|$ balances the losses at different microphones and across training mixtures.

According to the discussion in Section 3, minimizing $\mathcal{L}_{\text{MC}}$ would encourage separation of speakers. We illustrate the loss surface of $\mathcal{L}_{\text{MC}}$ in Appendix B.

Compared to the mixture consistency term proposed in [59], our mixture-constraint loss has very different physical meanings and mathematical forms. See Appendix E for detailed discussions.

### 4.3 FCP for relative RIR estimation

To compute $\mathcal{L}_{\text{MC}}$, we need to first estimate each of the relative RIRs, $\hat{\mathbf{g}}_p(c,f)$. In [29, 60], FCP is proposed to estimate the relative RIR relating direct-path signal to reverberant image for speech dereverberation. In this study, we employ FCP to estimate the relative RIR relating $\hat{Z}(c)$ to the speaker image captured at each microphone $p$ (i.e., $X_p(c)$).

Assuming speakers are non-moving, we estimate relative RIRs by solving the following problem:

$$\hat{\mathbf{g}}_p(c,f) = \operatorname*{argmin}_{\mathbf{g}_p(c,f)} \sum_t \frac{1}{\hat{\lambda}_p(c,t,f)} \Big| Y_p(t,f) - \mathbf{g}_p(c,f)^{\mathsf{H}} \widetilde{\mathbf{Z}}(c,t,f) \Big|^2, \tag{6}$$

where $\mathbf{g}_p(c,f) \in \mathbb{C}^{I+1+J}$ is a $K$-tap (with $K = I+1+J$) time-invariant FCP filter, $\widetilde{\mathbf{Z}}(c,t,f) = [\hat{Z}(c,t-I,f),\ldots,\hat{Z}(c,t,f),\ldots,\hat{Z}(c,t+J,f)]^{\mathsf{T}} \in \mathbb{C}^K$ stacks $I$ past and $J$ future T-F units with the current one. Since the actual number of filter taps (i.e., $A$ and $B$ defined in the text below (2)) is unknown, we set them to $I$ and $J$, both of which are hyper-parameters to tune. $\hat{\lambda}_p(c,t,f)$ is a weighting term balancing the importance of each T-F unit. Following [29], we define it as $\hat{\lambda}_p(c,t,f) = \left(\frac{1}{P}\sum_{p'=1}^P |Y_{p'}(t,f)|^2\right) + \xi \times \max\left(\frac{1}{P}\sum_{p'=1}^P |Y_{p'}|^2\right)$, where $\xi$ ($= 10^{-4}$ in this study) is used to floor the weighting term and $\max(\cdot)$ extracts the maximum value of a spectrogram. (6) is a weighted linear regression problem. A closed-form solution can be readily computed:

$$\hat{\mathbf{g}}_p(c,f) = \Big(\sum_t \frac{1}{\hat{\lambda}_p(c,t,f)} \widetilde{\mathbf{Z}}(c,t,f)\widetilde{\mathbf{Z}}(c,t,f)^{\mathsf{H}}\Big)^{-1} \sum_t \frac{1}{\hat{\lambda}_p(c,t,f)} \widetilde{\mathbf{Z}}(c,t,f)(Y_p(t,f))^*, \tag{7}$$

where $(\cdot)^*$ computes complex conjugate. We then plug $\hat{\mathbf{g}}_p(c,f)$ into (4) and compute the loss.

Note that to compute the relative RIR, ideally we should filter $\hat{Z}(c)$ to approximate $X_p(c)$ (i.e., replacing $Y_p$ in (6) with $X_p(c)$), but $X_p(c)$ is unknown. In (6), we instead linearly filter $\hat{Z}(c)$ to approximate $Y_p$, and earlier studies [29, 60] suggest that the resulting $\hat{\mathbf{g}}_p(c,f)^{\mathsf{H}} \widetilde{\mathbf{Z}}(c,t,f)$ would be an estimate of $X_p(c,t,f)$, if $\hat{Z}(c)$ is reasonably accurate (see Appendix C for the derivation). We name the speaker image estimated this way as *FCP-estimated image*:

$$\hat{X}_p^{\text{FCP}}(c,t,f) = \hat{\mathbf{g}}_p(c,f)^{\mathsf{H}} \widetilde{\mathbf{Z}}(c,t,f). \tag{8}$$

It is therefore reasonable to sum up the FCP-estimated images of all the speakers and define a loss between the summation and $Y_p$ as in (4).

Although (6) appears similar to multi-channel linear prediction (MCLP) [61, 62] which is popular in conventional speech separation algorithms, we point out that they have very different physical meanings. We consider that (6) does *forward filtering*, where source estimates are filtered to approximate mixtures so that relative RIRs can be estimated, while MCLP does *inverse filtering*, where mixtures are filtered to approximate target sources and the filters are designed to suppress non-target signals. This difference results in non-trivial changes of the physical meanings of the computed filters (see also discussions in Section V.C of [29]).

Although there were earlier efforts in estimating relative RIRs [56], they are based on conventional signal processing techniques and the performance is usually limited due to strong assumptions on, and inaccurate estimation of, signal statistics.

### 4.4 Time alignment issues and alternative loss functions

In (4), we do not filter the DNN estimates when computing the loss on the first (reference) microphone. We expect this to result in a $\hat{Z}(c)$ time-aligned with the speaker image $X_1(c)$ (i.e., $\hat{Z}(c)$ is an estimate

of $X_1(c)$). Since the reference microphone may not be the microphone closest to speaker $c$, it is best to use non-causal filters when filtering $\hat{Z}(c)$ to approximate the reverberant image $X_p(c)$ at non-reference microphones that are closer to source $c$ than the reference microphone, and instead use causal filters for non-reference microphones that are farther[2]. Since estimating which non-reference microphones are closer or farther to a source than the reference microphone is not an easy task and doing this would complicate our system, we can just choose to use non-causal filters for all the non-reference microphones. This could, however, limit the DNN's capability at separating the speakers, because the relative RIRs for some non-reference microphones (farther to source $c$ than the reference microphone) are causal, and it may not be a good idea to assume non-causal filters.

To address this issue, we make a simple modification to the loss function in (4):

$$\mathcal{L}_{\text{MC}} = \sum_{p=1}^{P} \alpha_p \mathcal{L}_{\text{MC},p} = \sum_{p=1}^{P} \alpha_p \sum_{t,f} \mathcal{F}\Big(Y_p(t,f), \sum_{c=1}^{C} \hat{\mathbf{g}}_p(c,f)^{\mathsf{H}} \, \widetilde{\mathbf{Z}}(c,t,f)\Big), \quad (9)$$

where the difference is that we also filter the DNN estimates when computing the loss on the reference microphone, and we constrain $\hat{\mathbf{g}}_p(c,f)$ to be causal and that $\widetilde{\mathbf{Z}}(c,t,f)$ only stacks current and past frames. This way, the resulting $\hat{Z}(c)$ would not be time-aligned with the revererberant image captured at the reference microphone (i.e., $X_1(c)$) or any other non-reference microphones. Because of the causal filtering, $\hat{Z}(c)$ would be more like an estimate of the reverberant image captured by a *virtual microphone* that is closer to speaker $c$ than all the $P$ microphones (see Appendix G.1 for an interpretation). It would contain less reverberation of speaker $c$ than any of the speaker images captured by the $P$ microphones due to the causal filtering.

To produce an estimate that is time-aligned with the reverberant image at a microphone (e.g., $X_p(c)$), we use the FCP-estimated image computed in (8) (i.e., $\hat{X}_p^{\text{FCP}}(c)$) as the output.

### 4.5 Addressing frequency permutation problem

In (4) and (9), FCP is performed in each frequency independently from the others. Even though the speakers are separated at each frequency, the separation results of the same speaker at different frequencies may however not be grouped into the same output spectrogram (see an example in Appendix D). This is known as the *frequency permutation problem* [41], which has been studied for decades in frequency-domain blind source separation algorithms such as frequency-domain independent component analysis [37–41] and spatial clustering [45–48]. Popular solutions for frequency alignment are designed by leveraging cross-frequency correlation of spectral patterns [47, 64] and direction-of-arrival estimation [65]. However, these solutions are often empirical and have a complicated design. They can be used to post-process DNN estimates for frequency alignment, but it is not easy to integrate them with UNSSOR for joint training. This section proposes a loss term, with which the trained DNN can learn to produce target estimates without frequency permutation.

To deal with frequency permutation, IVA [42–44] assumes that, at each frame, the de-mixed outputs at all the frequencies follow a complex Gaussian distribution with a shared variance term across frequencies: $\mathbf{w}(c,f)^{\mathsf{H}}\mathbf{Y}(t,f) \sim \mathcal{N}(0, D(t,c))$, where $\mathbf{w}(c,f) \in \mathbb{C}^P$ is the de-mixing weight vector (in a time-invariant de-mixing matrix) for speaker $c$ at frequency $f$, and $D(t,c) \in \mathbb{R}$ is the shared variance term, which is assumed time-variant. When maximum likelihood estimation is performed to estimate the de-mixing matrix, the variance term shared across all the frequencies is found very effective at solving the frequency permutation problem [41–44].

Motivated by IVA, we design the following loss term, named *intra-source magnitude scattering* (ISMS), to alleviate the frequency permutation problem in DNN outputs:

$$\mathcal{L}_{\text{ISMS}} = \sum_{p=1}^{P} \alpha_p \mathcal{L}_{\text{ISMS},p} = \sum_{p=1}^{P} \alpha_p \frac{\sum_t \frac{1}{C} \sum_{c=1}^{C} \text{var}\Big(\log(|\hat{X}_p^{\text{FCP}}(c,t,\cdot)|)\Big)}{\sum_t \text{var}\Big(\log(|Y_p(t,\cdot)|)\Big)}, \quad (10)$$

where $\hat{X}_p^{\text{FCP}}$ is computed via (8), $\hat{X}_p^{\text{FCP}}(c,t,\cdot) \in \mathbb{C}^F$, and $\text{var}(\cdot)$ computes the variance of the values in a vector. At each frame, we essentially want the the magnitudes of the estimated spectrogram of each speaker (i.e., $\hat{X}_p^{\text{FCP}}(c,t,\cdot)$) to have a small intra-source variance. The rationale is that, when

---

[2]Note that the relative RIR relating a signal to its delayed version is causal and the relative RIR relating a signal to its advanced version is non-causal [63]. See Appendix F for an intuitive explanation.

frequency permutation happens, $\hat{X}_p^{\text{FCP}}(c, t, \cdot)$ would contain multiple sources, and the resulting variance would be larger than that computed when $\hat{X}_p^{\text{FCP}}(c, t, \cdot)$ contains only one source. $\mathcal{L}_{\text{ISMS}}$ echoes IVA's idea of assuming a shared variance term across all the frequencies. If the ratio in (10) becomes smaller, it indicates that the magnitudes of $\hat{X}_p^{\text{FCP}}(c, t, \cdot)$ are more centered around their mean. This is similar to optimizing the likelihood of $\hat{X}_p^{\text{FCP}}(c, t, \cdot)$ under a Gaussian distribution with a variance term shared across all the frequencies. In (10), a logarithmic compression is applied, since log-compressed magnitudes better follow Gaussian distribution than raw magnitudes [66].

We combine $\mathcal{L}_{\text{ISMS}}$ with $\mathcal{L}_{\text{MC}}$ in (9) for DNN training, using a weighting term $\gamma \in \mathbb{R}$:

$$\mathcal{L}_{\text{MC+ISMS}} = \mathcal{L}_{\text{MC}} + \gamma \times \mathcal{L}_{\text{ISMS}}. \tag{11}$$

### 4.6 Training UNSSOR for monaural unsupervised separation

UNSSOR can be trained for monaural unsupervised separation by only feeding the mixture at the reference microphone to the DNN but still computing the loss on multiple microphones. Fig. 1 illustrates the idea. At run time, the trained system performs monaural under-determined separation, and multi-microphone over-determined mixtures are only required for DNN training. The loss computed at multiple microphones could guide the DNN to exploit monaural spectro-temporal patterns for separation, even in an unsupervised setup.

## 5 Experimental setup

We validate the proposed algorithms on two-speaker separation in reverberant conditions based on the six-channel SMS-WSJ dataset [67] (see Appendix A for its details). This section describes the baseline systems and evaluation setup. See Appendix I for miscellaneous system and DNN setup.

### 5.1 Baselines

The baselines include conventional unsupervised separation algorithms, an improved version of RAS, MixIT, and supervised learning based models.

We include spatial clustering [46–48] for comparison. We use a public implementation[3][68], which leverages complex angular-central Gaussian mixture models [48] for sub-band spatial clustering and exploits inter-frequency correlation of cluster posteriors [46, 64] for frequency alignment. The number of sources is set to three, one of which is used for garbage collection, following [44]. After obtaining the estimates, we discard the one with the lowest energy. The STFT window size is tuned to 128 ms and hop size to 16 ms.

We include IVA [41, 44] for comparison. We use the public implementations provided by the *torchiva* toolkit [69]. We use the default spherical Laplacian model to model source distribution. In over-determined cases, the number of sources is set to three and we discard the estimate with the lowest energy, similarly to the setup in the spatial clustering baseline[4]. The STFT window size is tuned to 256 ms and hop size to 32 ms.

We propose a novel variant of the RAS algorithm [36] for comparison. Appendix H discusses the differences between UNSSOR and RAS. Since RAS cannot achieve unsupervised separation, we improve it by computing loss on multi-microphone mixtures, and name the new algorithm as improved RAS (iRAS). We employ the time-domain Wiener filtering (WF) technique in [36] to filter re-synthesized time-domain estimates $\hat{z}(c) = \text{iSTFT}(\hat{Z}(c))$, where $\hat{Z}(c)$ is produced by TF-GridNet. The loss is defined as:

$$\mathcal{L}_{\text{iRAS}} = \sum_{p=1}^{P} \alpha_p \mathcal{L}_{\text{iRAS},p} = \sum_{p=1}^{P} \alpha_p \frac{1}{\|y_p\|_1} \Big\| y_p - \sum_{c=1}^{C} \hat{h}_p(c) * \hat{z}(c) \Big\|_1, \tag{12}$$

with $*$ denoting linear convolution, $y_p$ the time-domain mixture at microphone $p$, and $\hat{h}_p(c)$ a time-domain Wiener filter computed by solving the following problem:

$$\hat{h}_p(c) = \text{argmin}_{h_p(c)} \|y_p - h_p(c) * \hat{z}(c)\|_2^2, \tag{13}$$

---

[3] https://github.com/fgnt/pb_bss/blob/master/examples/mixture_model_example.ipynb
[4] For IVA and spatial clustering, using a garbage source leads to better separation in our experiments.

Table 1: Averaged results of 2-speaker separation on SMS-WSJ (6-channel input and loss).

| Row | Systems | $I$ | $J$ | Loss | Val. set SDR (dB) | Test set SDR (dB) | Test set SI-SDR (dB) | Test set PESQ | Test set eSTOI |
|---|---|---|---|---|---|---|---|---|---|
| 0a | Mixture | - | - | - | 0.1 | 0.1 | 0.0 | 1.87 | 0.603 |
| 1a | UNSSOR | 19 | 0 | $\mathcal{L}_{\text{MC}}$ | 12.5 | 11.9 | 10.2 | 2.61 | 0.735 |
| 1b | UNSSOR + Corr. based freq. align. | 19 | 0 | $\mathcal{L}_{\text{MC}}$ | **16.1** | **15.7** | **14.7** | **3.47** | 0.884 |
| 1c | UNSSOR + Oracle freq. align. | 19 | 0 | $\mathcal{L}_{\text{MC}}$ | 16.2 | 15.8 | 14.9 | 3.48 | 0.889 |
| 2a | UNSSOR | 19 | 0 | $\mathcal{L}_{\text{MC+ISMS}}$ | 16.0 | 15.6 | 14.6 | 3.44 | **0.885** |
| 2b | UNSSOR + Corr. based freq. align. | 19 | 0 | $\mathcal{L}_{\text{MC+ISMS}}$ | 16.0 | 15.6 | **14.7** | 3.44 | **0.885** |
| 2c | UNSSOR + Oracle freq. align. | 19 | 0 | $\mathcal{L}_{\text{MC+ISMS}}$ | 16.0 | 15.6 | 14.7 | 3.44 | 0.886 |
| 3a | Spatial clustering + Corr. based freq. align. [44] | - | - | - | 8.8 | 8.6 | 7.4 | 2.44 | 0.726 |
| 3b | IVA [44] | - | - | - | 10.3 | 10.6 | 8.9 | 2.58 | 0.764 |
| 3c | iRAS w/ causal 512-tap filters | - | - | $\mathcal{L}_{\text{iRAS}}$ | 7.8 | 7.6 | 5.7 | 2.14 | 0.642 |
| 3d | iRAS w/ non-causal 512-tap filters (100 future taps) | - | - | $\mathcal{L}_{\text{iRAS}}$ | 8.0 | 7.8 | 5.7 | 2.13 | 0.637 |
| 4a | PIT (supervised) [23] | - | - | - | 19.9 | 19.4 | 18.9 | 4.08 | 0.949 |

*Notes*: the rows shows in grey indicate using oracle information or using supervised models.

Table 2: Averaged results of 2-speaker separation on SMS-WSJ (3-channel input and loss).

| Row | Systems | $I$ | $J$ | Loss | Val. set SDR (dB) | Test set SDR (dB) | Test set SI-SDR (dB) | Test set PESQ | Test set eSTOI |
|---|---|---|---|---|---|---|---|---|---|
| 0a | Mixture | - | - | - | 0.1 | 0.1 | 0.0 | 1.87 | 0.603 |
| 1a | UNSSOR | 19 | 0 | $\mathcal{L}_{\text{MC}}$ | 9.9 | 9.4 | 7.4 | 2.12 | 0.672 |
| 1b | UNSSOR + Corr. based freq. align. | 19 | 0 | $\mathcal{L}_{\text{MC}}$ | 15.3 | 15.0 | 13.9 | 3.18 | 0.867 |
| 1c | UNSSOR + Oracle freq. align. | 19 | 0 | $\mathcal{L}_{\text{MC}}$ | 15.5 | 15.2 | 14.1 | 3.19 | 0.871 |
| 2a | UNSSOR | 19 | 0 | $\mathcal{L}_{\text{MC+ISMS}}$ | **15.7** | **15.4** | **14.4** | **3.20** | 0.874 |
| 2b | UNSSOR + Corr. based freq. align. | 19 | 0 | $\mathcal{L}_{\text{MC+ISMS}}$ | **15.7** | **15.4** | **14.4** | **3.20** | **0.875** |
| 2c | UNSSOR + Oracle freq. align. | 19 | 0 | $\mathcal{L}_{\text{MC+ISMS}}$ | 15.8 | 15.4 | 14.5 | 3.20 | 0.876 |
| 3a | Spatial clustering + Corr. based freq. align. [44] | - | - | - | 9.6 | 9.5 | 8.5 | 2.52 | 0.759 |
| 3b | IVA [44] | - | - | - | 11.6 | 12.0 | 10.7 | 2.67 | 0.802 |
| 3c | iRAS w/ causal 512-tap filters | - | - | $\mathcal{L}_{\text{iRAS}}$ | 5.1 | 4.8 | 2.7 | 1.88 | 0.588 |
| 3d | iRAS w/ non-causal 512-tap filters (100 future taps) | - | - | $\mathcal{L}_{\text{iRAS}}$ | 4.6 | 4.5 | 2.2 | 1.87 | 0.579 |
| 4a | PIT (supervised) [23] | - | - | - | 17.4 | 16.8 | 16.3 | 3.91 | 0.924 |

which is quadratic and has a closed-form solution. The separation result is computed as $\hat{x}_p^{\text{WF}}(c) = \hat{h}_p(c) * \hat{z}(c)$. Following [36], we use 512 filter taps, and filter the future 100, the current, and the past 411 samples (i.e., non-causal filtering). We can also filter the current and the past 511 samples (i.e., causal filtering), and additionally experiment with a filter length (in time) same as the length of the FCP filters (see Appendix J).

For comparison, we also include MixIT [30], which requires using synthetic mixtures of mixtures.

We report the result of using supervised learning, where PIT [5] is used to address the permutation problem. This result can be viewed as a performance upper bound of unsupervised separation.

We use the same DNN and training configurations as those in UNSSOR for a fair comparison.

## 5.2 Evaluation setup and metrics

We designate the first microphone as the reference microphone, and use the time-domain signal corresponding to $X_1(c)$ of each speaker $c$ for metric computation. The evaluation metrics include signal-to-distortion ratio (SDR) [70], scale-invariant SDR (SI-SDR) [71], perceptual evaluation of speech quality (PESQ) [72], and extended short-time objective intelligibility (eSTOI) [73]. SI-SDR and SDR evaluate the sample-level accuracy of predicted signals, and PESQ and eSTOI are objective metrics of speech quality and intelligibility respectively. For all the metrics, the higher, the better.

## 6 Evaluation results

This section reports evaluation results on SMS-WSJ and compares the performance of various setups.

### 6.1 Effectiveness of UNSSOR at promoting separation

Table 1 and 2 respectively report the results of using six- and three-microphone input and loss. After hyper-parameter tuning, in default we use the loss in (9) for DNN training, set $I = 19$ and $J = 0$ (defined below (6)) for FCP (i.e., causal FCP filtering with 20 taps), and set $\alpha_p = 1$ (meaning no

Table 3: Averaged results of 2-speaker separation on SMS-WSJ (1-channel input and 6-channel loss).

| Row | Systems | $I$ | $J$ | Loss | Val. set SDR (dB) | Test set SDR (dB) | SI-SDR (dB) | PESQ | eSTOI |
|-----|---------|-----|-----|------|--------|--------|---------|------|-------|
| 0a | Mixture | - | - | - | 0.1 | 0.1 | 0.0 | 1.87 | 0.603 |
| 1a | UNSSOR | 19 | 1 | $\mathcal{L}_{\text{MC+ISMS}}$ | **13.0** | **12.5** | **11.9** | **3.27** | **0.832** |
| 2a | iRAS w/ causal 512-tap filters | - | - | $\mathcal{L}_{\text{iRAS}}$ | 7.5 | 7.2 | 5.6 | 2.03 | 0.641 |
| 2b | iRAS w/ non-causal 512-tap filters (100 future taps) | - | - | $\mathcal{L}_{\text{iRAS}}$ | 10.7 | 10.5 | 9.7 | 2.80 | 0.778 |
| 3a | Monaural PIT (supervised) [23] | - | - | - | 16.2 | 15.7 | 15.3 | 3.79 | 0.907 |

Table 4: Averaged results of 2-speaker separation on SMS-WSJ (1-channel input and 3-channel loss).

| Row | Systems | $I$ | $J$ | Loss | Val. set SDR (dB) | Test set SDR (dB) | SI-SDR (dB) | PESQ | eSTOI |
|-----|---------|-----|-----|------|--------|--------|---------|------|-------|
| 0a | Mixture | - | - | - | 0.1 | 0.1 | 0.0 | 1.87 | 0.603 |
| 1a | UNSSOR | 19 | 1 | $\mathcal{L}_{\text{MC+ISMS}}$ | **12.5** | **12.0** | **11.4** | **3.18** | **0.822** |
| 2a | iRAS w/ causal 512-tap filters | - | - | $\mathcal{L}_{\text{iRAS}}$ | −0.1 | −0.3 | −3.0 | 1.62 | 0.453 |
| 2b | iRAS w/ non-causal 512-tap filters (100 future taps) | - | - | $\mathcal{L}_{\text{iRAS}}$ | 11.0 | 10.7 | 9.9 | 2.81 | 0.783 |
| 3a | Monaural PIT (supervised) [23] | - | - | - | 16.2 | 15.7 | 15.3 | 3.79 | 0.907 |

weighting is applied for different microphones). For the 3-microphone case, we use the mixtures at the first, third, and fifth microphones for training and testing. Notice that for two-speaker separation, the cases with six or three microphones are both over-determined.

In both tables, from row 1a we observe that UNSSOR produces reasonable separation of speakers, improving the SDR from 0.1 to, for example, 12.5 dB in Table 1, but its output suffers from the frequency permutation problem (see Appendix D for an example). In row 1c, we use oracle target speech to obtain oracle frequency alignment and observe much better results over 1a. This shows the effectiveness of $\mathcal{L}_{\text{MC}}$ at promoting separation of speakers and the severity of the frequency permutation problem. In row 1b, we use a frequency alignment algorithm (same as that used in the spatial clustering baseline) [46, 64] to post-process the separation results of 1a. This algorithm leads to impressive frequency alignment (see 1b vs. 1c), but it is empirical and has a complicated design.

## 6.2 Effectiveness of ISMS loss at addressing frequency permutation problem

We train DNNs using $\mathcal{L}_{\text{MC+ISMS}}$ defined in (11). In each case (i.e, six- and three-microphone), we separately tune the weighting term $\gamma$ in (11) based on the validation set. In both table, comparing row 2a-2c with 1a-1c, we observe that including $\mathcal{L}_{\text{ISMS}}$ is very effective at dealing with the frequency permutation problem, yielding almost the same performance as using oracle frequency alignment.

## 6.3 Results of training UNSSOR for monaural unsupervised separation

Table 3 and 4 use the mixture only at the reference microphone 1 as the network input, while computing the loss respectively on three and six microphones. We tune $J$ to 1 (i.e., non-causal FCP filter), considering that, for a specific target speaker, the reference microphone may not be the microphone closest to that speaker[5]. See an interpretation on why non-causal filtering is needed in Appendix G.2. We still set the microphone weight $\alpha_p$ to 1.0 for non-reference microphones (i.e., when $p \neq 1$), but tune $\alpha_1$ to a smaller value based on the validation set. Without using a smaller $\alpha_1$, we found that the DNN easily overfits to microphone 1, as we use the mixture at microphone 1 as the only input and compute the MC loss also on the mixture at microphone 1. The DNN can just aggressively optimize $\mathcal{L}_{\text{MC},p}$ to zero at microphone 1 and not optimize that at other microphones.

From row 1a of both tables, strong performance is observed in this under-determined setup, indicating that the multi-microphone loss can inform the DNN what desired target sound objects are and the DNN can learn to model spectral patterns in speech for unsupervised separation. In addition, the result in 1a of Table 3 is better than that in Table 4. This indicates that using more microphones as constraints in the proposed MC loss can elicit better separation.

---

[5]We do not need many future taps, considering that the hop size is 8 ms in our system and the microphone array in SMS-WSJ is a compact array with a diameter of 20 cm. In air, sound would travel $340 \times 0.008 = 2.72$ meters in 8 ms if its speed is 340 meters per second. This distance is far larger than the array aperture size.

### 6.4 Comparison with other methods and supplementary results

In Table 1-4, we compare the performance of UNSSOR with spatial clustering, IVA, iRAS, and supervised PIT-based models. In Appendix J, we compare UNSSOR and iRAS when they use the same filter length (in time). UNSSOR shows better performance than previous unsupervised separation models that can be performed or trained directly on mixtures. A sound demo is available at this link[6]. UNSSOR is worse than supervised PIT but the performance is reasonably strong. For example, in row 2a of Table 2, UNSSOR obtains 15.4 dB SDR on the test set, which is close to the 16.8 dB result obtained by supervised PIT in 4a. In Appendix L, we compare the results of UNSSOR with MixIT [30] and observe better performance.

We think that the effectiveness of the proposed system comes from both the strong modeling capability of TF-GridNet and the UNSSOR mechanism itself. In Appendix K, we use UNSSOR with other separation models such as TCN-DenseUNet [74] and DPRNN [75], which are known to have weaker modelling capability than TF-GridNet in supervised separation tasks (see for example [23]). We observe that the separation performance is worse but still reasonable.

## 7 Limitations

Our study shows the strong potential of UNSSOR for unsupervised speech separation. There are, however, several weaknesses we need to address in future research. First, we assume that sources are directional point sources so that each relative RIR can be modelled using a short filter, and diffuse sources are not considered. Second, we assume that sources are non-moving within each utterance so that we can use time-invariant FCP filters. Third, we assume that the number of sources is known and the sources are fully-overlapped. Fourth, only measurement or modeling noise is considered and realistic directional or diffuse background noises with strong energy are not included. Although these assumptions are also made in many algorithms such as IVA, spatial clustering, RAS and iRAS, they need to be addressed to realize more practical and robust speech separation systems.

## 8 Conclusion

We have proposed UNSSOR for unsupervised neural speech separation. We show that it is possible to train unsupervised models directly on mixtures, if the mixtures are over-determined. We have proposed mixture-constraint loss functions, which leverage multi-microphone mixtures as constraints, to promote separation of speakers. We find that minimizing ISMS can alleviate the frequency permutation problem. Although UNSSOR requires over-determined training mixtures, it can be trained to perform under-determined unsupervised separation. Future research will combine UNSSOR with semi-supervised learning, evaluate it on real-recorded noisy-reverberant data such as CHiME-6 [76], AMI [77] and AliMeeting [78], and address the limitations described in Section 7.

In closing, we emphasize that a key scientific contribution of this paper is that the over-determined property afforded by having more microphones than speakers can narrow down the solutions to the underlying sources, and this property can be leveraged to design a supervision to train DNNs to model speech patterns via unsupervised learning and realize unsupervised separation. This meta-idea, we believe, would motivate the design of many algorithms in future research on neural source separation.

### Acknowledgments and Disclosure of Funding

We would like to thank Dr. Robin Scheibler at LINE Corporation for constructive discussions on IVA, and Dr. Samuele Cornell for helpful discussions on MixIT. This research is part of the Delta research computing project, which is supported by the National Science Foundation (Award OCI 2005572) and the State of Illinois. Delta is a joint effort of the University of Illinois at Urbana-Champaign and its National Center for Supercomputing Applications. We also gratefully acknowledge the support of NVIDIA Corporation with the donation of the RTX 8000 GPUs used in this research.

---

[6]`https://zqwang7.github.io/demos/UNSSOR-demo/index.html`.

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

## Appendix

This appendix is organized as follows:

- Appendix A describes the SMS-WSJ dataset.
- Appendix B visualizes the surface of the proposed mixture-constraint loss to show that minimizing the loss can promote unsupervised separation of speakers.
- Appendix C provides the derivation that FCP can be used to approximate speaker images.
- Appendix D illustrates the frequency permutation problem.
- Appendix E describes differences between the MC loss and the mixture consistency concept.
- Appendix F illustrates the cases on when to use causal and non-causal filtering.
- Appendix G provides interpretations of the physical meanings of intermediate DNN estimates $\hat{Z}$.
- Appendix H discusses differences between UNSSOR and RAS.
- Appendix I presents miscellaneous system and DNN configurations.
- Appendix J experiments with alternative filters taps for iRAS and UNSSOR.
- Appendix K experiments UNSSOR with DNN architectures with lower modelling capabilities.
- Appendix L reports the results of MixIT.

## A    SMS-WSJ dataset

SMS-WSJ [67] is a popular corpus for evaluating two-speaker separation algorithms in reverberant conditions. The clean speech is sampled from the WSJ0 and WSJ1 datasets. The corpus contains $33,561$ ($\sim$87.4 h), $982$ ($\sim$2.5 h), and $1,332$ ($\sim$3.4 h) two-speaker mixtures respectively for training, validation, and testing. The simulated microphone array has six microphones arranged uniformly on a circle with a diameter of 20 cm. For each mixture, the speaker-to-array distance is drawn from the range $[1.0, 2.0]$ m, and the reverberation time (T60) is sampled from $[0.2, 0.5]$ s. A weak white noise is added to simulate microphone self-noises, and the energy level between the sum of the reverberant speech signals and the noise is sampled from the range $[20, 30]$ dB. The sampling rate is 8 kHz.

## B    Visualization of loss surface

Fig. 2 visualizes the values of $\mathcal{L}_{\text{MC}}$ in (4) based on a six-channel noisy-reverberant two-speaker mixture sampled from the SMS-WSJ dataset (see Appendix A for the dataset details). Let $C = 2$ and suppose that the DNN estimates are

$$\hat{Z}(1) = \mu \times X_1(1) + \nu \times X_1(2) + \varepsilon_1/2$$
$$\hat{Z}(2) = (1 - \mu) \times X_1(1) + (1 - \nu) \times X_1(2) + \varepsilon_1/2, \tag{14}$$

where $\mu$ and $\nu \in \mathbb{R}$ are bounded in the range $[0, 1]$. Essentially, we use $\mu$ and $\nu$ to mimic the cases that the DNN produces (1) good separation (i.e., when $\mu \approx 1$ and $\nu \approx 0$, or $\mu \approx 0$ and $\nu \approx 1$); and (2) bad separation (i.e., when $\mu$ and $\nu$ are both away from 0 and 1, meaning that each estimate contains multiple speakers, and when $\mu \approx 0$ and $\nu \approx 0$, or $\mu \approx 1$ and $\nu \approx 1$, meaning that the two speakers are merged into one output and the other output does not contain any speakers). Fig. 2 enumerates $\mu$ and $\nu$ and plots the resulting separation results against the loss value of (4). We can see the loss values are smallest when $\mu \approx 1$ and $\nu \approx 0$ or when $\mu \approx 0$ and $\nu \approx 1$ (i.e., when the speakers are successfully separated), and clearly larger otherwise. This indicates that minimizing the proposed loss function can encourage separation.

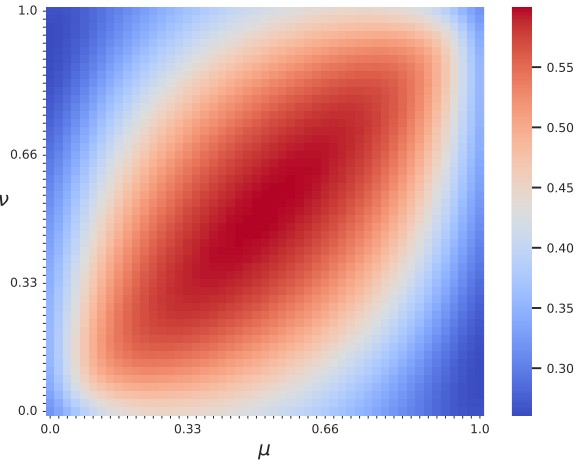

Figure 2: Loss surface of $\mathcal{L}_{\text{MC}}$ in (4) with $P = 6$ and $C = 2$ (i.e., over-determined conditions) against hypothesized separation outputs generated by using various $\mu$ and $\nu$. Best viewed in color.

## C  Effectiveness of FCP at approximating speaker images

Following the derivation in [29], let us define the mixture as $Y_p = X_p(c) + V_p(c)$, where $V_p(c)$ consists of the signals of all the sources but $c$. We can formulate (6) as

$$\underset{\mathbf{g}_p(c,f)}{\arg\min} \sum_t \frac{1}{\hat{\lambda}_p(c,t,f)} \left| X_p(c,t,f) + V_p(c,t,f) - \mathbf{g}_p(c,f)^{\mathsf{H}} \widetilde{\hat{\mathbf{Z}}}(c,t,f) \right|^2$$

$$\approx \underset{\mathbf{g}_p(c,f)}{\arg\min} \sum_t \frac{1}{\hat{\lambda}_p(c,t,f)} \left| X_p(c,t,f) - \mathbf{g}_p(c,f)^{\mathsf{H}} \widetilde{\hat{\mathbf{Z}}}(c,t,f)|^2 + |V_p(c,t,f) \right|^2$$

$$= \underset{\mathbf{g}_p(c,f)}{\arg\min} \sum_t \frac{1}{\hat{\lambda}_p(c,t,f)} \left| X_p(c,t,f) - \mathbf{g}_p(c,f)^{\mathsf{H}} \widetilde{\hat{\mathbf{Z}}}(c,t,f) \right|^2, \tag{15}$$

where the derivation from the first row to the second is based on the assumption that, as the DNN training continues, $\hat{Z}(c)$ would become more and more accurate so that, after some epochs, it can become uncorrelated (or little correlated) with $V_p(c)$, meaning that the cross-term would be small:

$$\sum_t \frac{1}{\hat{\lambda}_p(c,t,f)} \left( X_p(c,t,f) - \mathbf{g}_p(c,f)^{\mathsf{H}} \widetilde{\hat{\mathbf{Z}}}(c,t,f) \right)^{\mathsf{H}} V_p(c,t,f) \approx 0. \tag{16}$$

From the third row of (15), the resulting $\hat{\mathbf{g}}_p(c,f)^{\mathsf{H}} \widetilde{\hat{\mathbf{Z}}}(c,t,f)$ would approximate $X_p(c,t,f)$.

## D  Illustration of frequency permutation problem

We use three-channel input and loss for DNN training. Fig. 3(a) and (b) show an example separation result of the model trained with $\mathcal{L}_{\mathrm{MC}}$ in (9). Comparing the separated speech in Fig. 3(a) and (b) with the clean speech in (c) and (d), we can see that the separated speech suffers from the frequency permutation problem approximately in the range $[1.6, 2.9]$ kHz. Fig. 3(e) and (f) show the separation result of the model trained with $\mathcal{L}_{\mathrm{MC+ISMS}}$ in (11), which effectively addresses the frequency permutation problem.

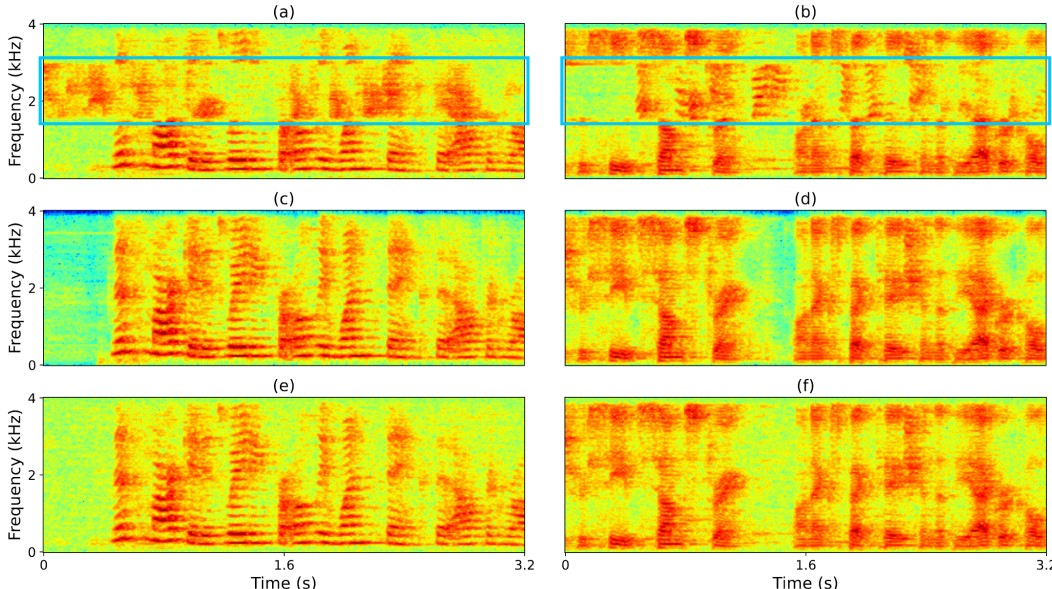

Figure 3: Example spectrograms of (a)-(b): FCP-estimated speaker image 1 and 2 with SDR scores of 8.7 and 7.7 dB (using $\mathcal{L}_{\mathrm{MC}}$ in (9) for training); (c)-(d): oracle speaker image 1 and 2; and (e)-(f): FCP-estimated speaker image 1 and 2 with SDR scores of 17.1 and 16.8 dB (using $\mathcal{L}_{\mathrm{MC+ISMS}}$ in (11) for training). The blue rectangles in (a) and (b) mark the region with frequency permutation. The mixture SDR scores of the two speakers are respectively 0.2 and $-0.1$ dB. Best viewed in color.

## E   Differences between mixture-constraint loss and mixture consistency

The proposed mixture-constraint loss, $\mathcal{L}_{\text{MC}}$, has very different physical meanings and mathematical forms compared to the *mixture consistency* term proposed in [59]. First, in [59], DNN estimates are strictly constrained to add up to the mixture (see Eq. (7) and (9) in [59]), while our MC loss only *encourages* the filtered DNN estimates to add up to the mixture. Second, our MC loss is applied to filtered DNN estimates rather than directly to DNN estimates. Third, we deal with multi-microphone MC, while [59] only addresses single-channel cases. Fourth, the motivation of the proposed MC loss is to use mixtures as constraints to regularize DNN estimates so that the estimates can approximate source images. This motivation is completely different from that of the mixture consistency term.

## F   Illustration of using causal and non-causal filtering

This section illustrates that the relative RIR relating a signal to its delayed version is causal and the relative RIR relating a signal to its advanced version is non-causal.

In Fig. 4, suppose that the blue signal is the DNN estimate for speaker $c$, and the orange signal is speaker $c$'s image at another microphone, which is a delayed version (i.e., reaching the microphone later). To filter the blue signal to approximate the oracle one, we only need a causal filter to delay the blue signal.

Reversely, suppose that the orange signal is the DNN estimate for speaker $c$, and the blue signal is speaker $c$'s image at another microphone, which is an advanced version (i.e., reaching the microphone earlier). To filter the orange signal to approximate the blue signal, we need a non-causal filter to advance the orangle signal.

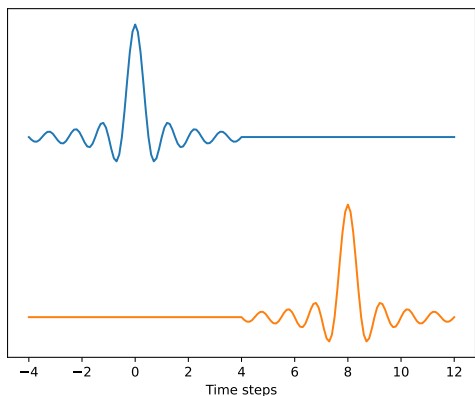

Figure 4: Example for illustrating causal and non-causal filtering. Best viewed in color.

## G   Interpretation of intermediate DNN estimates $\hat{Z}$

This section provides an interpretation of the physical meanings of the intermediate DNN estimates $\hat{Z}$ in the cases of using multi-channel input and loss, and single-channel input and multi-channel loss.

### G.1   Multi-channel-input case

In (9), $\hat{Z}(c)$ is constrained such that it can be filtered by a causal filter $\hat{\mathbf{g}}_p(c)$ to approximate $X_p(c)$. Since there could be an infinite number of combinations of $\hat{Z}(c)$ and $\hat{\mathbf{g}}_p(c)$ whose convolution results would well approximate $X_p(c)$, $\hat{Z}(c)$ cannot be interpreted as the dry source signal and we think it more similar to a signal captured by a virtual microphone that is closer to speaker $c$ than all the $P$ microphones. See Fig. 5(a) for an example, where each virtual microphone captures the direct-path signal of a target speaker earlier than any other microphones so that we can use causal FCP filters.

### G.2   Monaural-input case

In the monaural-input case, each $\hat{Z}(c)$ would be aligned to the speaker's image at the input microphone (i.e., the reference microphone), since the DNN only has monaural input and in this case the DNN is not likely to align its outputs to a virtual microphone or any actual microphones.

We give an example in Figure 5(b), where the reference microphone captures speaker 2's direct-path signal later than all the other microphones. In this case, we just need to use non-causal FCP filters when filtering $\hat{Z}(c)$ (which is estimated based on the monaural signal at the reference microphone) to approximate speaker 2's images captured at the other microphones.

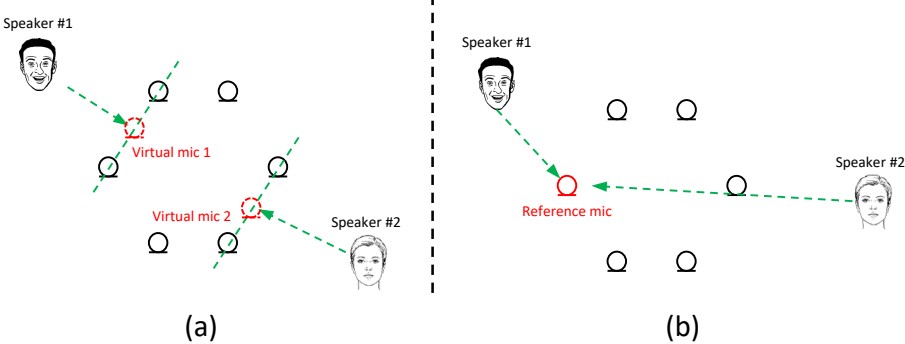

Figure 5: Illustration of signal alignment in (a) multi-channel-input, multi-channel-loss case; and (b) single-channel-input, multi-channel-loss case. Best viewed in color.

## H  Differences from RAS

In many aspects, UNSSOR differs from RAS [36], which deals with monaural two-speaker separation given binaural (two-channel) training mixtures. RAS first performs DNN-based monaural separation on the left-ear mixture in the magnitude T-F domain, then linearly filters the DNN estimates at the left ear through time-domain Wiener filtering such that the filtered estimate can approximate the right-ear mixture, and the training loss is computed between the summation of the filtered DNN estimates and the right-ear mixture. Besides differences in, for example, the DNN architectures, linear filtering algorithms, how phase estimation (important for estimating relative RIRs) is handled, how frequency permutation problem is dealt with, and training data curation (difficult training examples need to be removed to train RAS), the key difference is that RAS fails to be trained in an unsupervised way [36]. It still needs labelled mixtures so that a supervised PIT-based model can be trained and then used as the starting point for their unsupervised algorithm.

We think that the ineffectiveness of RAS in fully-unsupervised setup is likely because the loss is computed only on the right-ear mixture. Following our analysis in Section 3, in RAS there are $N \times 1$ equations (because the loss is only computed on the right-ear time-domain signal, assumed $N$-sample) but $N \times C + (2 - 1) \times 512 \times C$ unknowns (where the $512$ term is because the filter is assumed 512-tap in [36], and the $(2 - 1)$ term is because there is only one filter for each speaker in the binaural setup, i.e., only one non-reference microphone). This is an ill-posed problem, not likely to be solved via the current RAS algorithm.

## I  Miscellaneous system and DNN setup

In default, for STFT, the window size is 32 ms, the hop size is 8 ms, and the square-root Hann window is used as the analysis window. For $8$ kHz sampling rate, a $256$-point discrete Fourier transform is applied to extract 129-dimensional complex STFT spectra at each frame. This STFT setup is very common in modern deep learning based separation. Differently, the window and hop sizes of some baselines such as IVA and spatial clustering are considerably larger (see detailed setup in Section 5.1), as we observe that this leads to better separation performance, likely because more reverberation can be covered in each frame and, this way, their model assumptions can be better satisfied.

Our DNN architecture is TF-GridNet [23]. Using the symbols defined in Table I of [23], we set its hyper-parameters to $D = 48$, $B = 4$, $I = 4$, $J = 1$, $H = 192$, $L = 4$ and $E = 4$ for $8$ kHz sampling rate. Please do not confuse the symbols in TF-GridNet with the ones in this paper.

In each epoch, we sample an $l$-second mixture segment from each training mixture for model training. If a mixture is shorter than $l$ seconds, we pad zeros in the front rather than in the end, since padding in the end would result in a mixture that has abrupt stop of reverberation, which is not realistic and would be detrimental to FCP-based relative RIR estimation. In comparison, padding zeros in the front can avoid this problem. If a mixture is longer than $l$ seconds, we randomly pick an $l$-second segment. In default, $l$ is set to four seconds.

We normalize the sample variance of each sampled mixture segment to $1.0$, before feeding them to DNN for training. Adam (with the default setup in Pytorch v1.9) is used as the optimizer. The

Table 5: **Supplementary** averaged results of 2-speaker separation on SMS-WSJ (6-channel input and loss).

| Row | Systems | $I$ | $J$ | Loss | Val. set SDR (dB) | Test set SDR (dB) | Test set SI-SDR (dB) | Test set PESQ | Test set eSTOI |
|---|---|---|---|---|---|---|---|---|---|
| 2a | UNSSOR | 19 | 0 | $\mathcal{L}_{\text{MC+ISMS}}$ | 16.0 | 15.6 | 14.6 | 3.44 | 0.885 |
| 3e | iRAS w/ non-causal 1472-tap filters (64 future taps) | - | - | $\mathcal{L}_{\text{iRAS}}$ | 2.4 | 2.4 | 0.2 | 1.89 | 0.549 |

Table 6: **Supplementary** averaged results of 2-speaker separation on SMS-WSJ (3-channel input and loss).

| Row | Systems | $I$ | $J$ | Loss | Val. set SDR (dB) | Test set SDR (dB) | Test set SI-SDR (dB) | Test set PESQ | Test set eSTOI |
|---|---|---|---|---|---|---|---|---|---|
| 2a | UNSSOR | 19 | 0 | $\mathcal{L}_{\text{MC+ISMS}}$ | 15.7 | 15.4 | 14.4 | 3.20 | 0.874 |
| 3e | iRAS w/ non-causal 1472-tap filters (64 future taps) | - | - | $\mathcal{L}_{\text{iRAS}}$ | 7.7 | 7.2 | 5.4 | 1.87 | 0.621 |

Table 7: **Supplementary** averaged results of 2-speaker separation on SMS-WSJ (1-ch input and 6-ch loss).

| Row | Systems | $I$ | $J$ | Loss | Val. set SDR (dB) | Test set SDR (dB) | Test set SI-SDR (dB) | Test set PESQ | Test set eSTOI |
|---|---|---|---|---|---|---|---|---|---|
| 1a | UNSSOR | 19 | 1 | $\mathcal{L}_{\text{MC+ISMS}}$ | 13.0 | 12.5 | 11.9 | 3.27 | 0.832 |
| 2c | iRAS w/ non-causal 1536-tap filters (64 future taps) | - | - | $\mathcal{L}_{\text{iRAS}}$ | 1.7 | 1.7 | 0.0 | 1.90 | 0.561 |

Table 8: **Supplementary** averaged results of 2-speaker separation on SMS-WSJ (1-ch input and 3-ch loss).

| Row | Systems | $I$ | $J$ | Loss | Val. set SDR (dB) | Test set SDR (dB) | Test set SI-SDR (dB) | Test set PESQ | Test set eSTOI |
|---|---|---|---|---|---|---|---|---|---|
| 1a | UNSSOR | 19 | 1 | $\mathcal{L}_{\text{MC+ISMS}}$ | 12.5 | 12.0 | 11.4 | 3.18 | 0.822 |
| 2c | iRAS w/ non-causal 1536-tap filters (64 future taps) | - | - | $\mathcal{L}_{\text{iRAS}}$ | 1.3 | 1.3 | $-0.3$ | 1.87 | 0.549 |

$L_2$ norm for gradient clipping is set to $1.0$. The learning rate starts from $10^{-3}$ and is halved if the validation loss is not improved in two epochs. We terminate training once the learning rate is reduced to $6.25 \times 10^{-5}$. The batch size is set to four, with each segment being $4$-second long. For each model, an Nvidia A100 40GB GPU is used for training, and the model converges in three to four days.

We sweep $\gamma$ in (11) based on the set of $\{0.02, 0.04, 0.06, 0.1, 0.3, 1.0\}$. The microphone weight $\alpha_p$ in (4), (9), (10) and (12) is set to $1.0$ in default for all microphones (i.e., no weighting), and, in the monaural input case (e.g., in Table 3 and 4), we sweep $\alpha_1$ at the reference microphone 1 based on the set of $\{1/2, 1/3, 1/4, 1/5, 1/6, 1/8, 1/10\}$ to alleviate overfitting to microphone 1.

# J   Alternative filter length for iRAS and UNSSOR

We also provide the results of iRAS that uses the same filter length (in seconds) as that in UNSSOR. Given 8 kHz sampling rate, 32 ms window size, 8 ms hop size, and $K = I + 1 + J$ (defined below (6)) filter taps in FCP, we use $M = ((K - 1) \times 8 + 32)/1000 \times 8000$ filter taps for each $\hat{h}_p(c)$ in (12), and configure $\hat{h}_p(c)$ to filter the past $M - 8/1000 \times 8000 - 1$, the current, and the future 64 ($= 8/1000 \times 8000$) samples. We filter the future 8 ms of samples, because, in the STFT case, the hop size is 8 ms. We report the results in Table 5-8, each respectively corresponding to the results in Table 1-4. We observe that the performance is not as good as that of UNSSOR.

In Fig. 6, we make further comparisons of using different filter taps between UNSSOR and iRAS, following the experimental setup in the previous paragraph. Fig. 6(a) uses six-microphone input, Fig. 6(b) uses monaural input, and both of them use the six-microphone $\mathcal{L}_{\text{MC+ISMS}}$ (or $\mathcal{L}_{\text{iRAS}}$) loss. For UNSSOR, in Fig. 6(a) we set $J = 0$ and sweep $K \in \{5, 9, 13, 17, \mathbf{20}, 25\}$ and in Fig. 6(b) we set $J = 1$ and sweep $K \in \{5, 9, 13, 17, \mathbf{21}, 25\}$. For iRAS, we configure the filter to always filter the future 64 samples, and in Fig. 6(a) we sweep the filter taps $M \in \{128, 256, 384, 512, 768, 1024, 1280, \mathbf{1472}\}$ and in Fig. 6(b) we sweep $M \in \{128, 256, 384, 512, 768, 1024, 1280, \mathbf{1536}\}$. Notice that in Fig. 6, the filter taps $M$ in iRAS and $K$ in UNSSOR are vertically corresponding to each other. That is, some swepted filter taps $M$ are computed based on the swepted $K$ (e.g., for $M = 1024$ and $K = 13$, we have $1024 = ((13 - 1) \times 8 + 32)/1000 \times 8000)$ so that they can result in the same filter length in time. From the figures, we observe that the best filter length (in seconds) is different for UNSSOR and iRAS, and the best performance of UNSSOR is higher than that of iRAS.

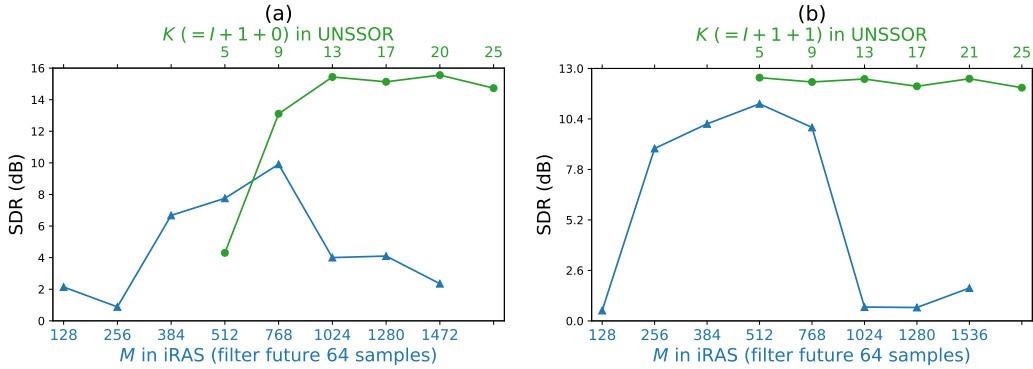

Figure 6: Averaged SDR (dB) results of UNSSOR and iRAS using various filter taps in cases of using (a) 6-channel input and loss; and (b) 1-channel input and 6-channel loss on SMS-WSJ. Vertically-corresponded $M$ and $K$ mean that the filter lengths in time are the same. Best viewed in color.

We point out that using a very long filter (i.e., a large $M$) for time-domain Wiener filtering would prevent separation, since, in that case, the filter would have a large degrees of freedom to filter $\hat{z}(c)$ to fit $y_p$ very well and obtain a small $\mathcal{L}_{\text{iRAS}}$ (see (13)), even though $\hat{z}(c)$ is not a good separation result. From Fig. 6(a), it can be observed that a very large $M$ (e.g., 1472) does not yield good separation; and in addition, a very small $M$ (e.g., 128) is also not good, as the linear filter could be just too short to fit the mixture $y_p$ well. Similar trend can also be observed in the results of UNSSOR. For example, in Fig. 6(a), setting $K$ to 5 and 25 produces worse performance than to 20. We can see that the filter length is an important hyper-parameter to tune.

We emphasize that, to compute the closed-form solution of time-domain Wiener filtering (see (13)), we need to invert a big $M \times M$ matrix for each mixture, while in FCP (see (6)), we only need to invert a much smaller $K \times K$ matrix for each of the $F$ frequencies. Using FCP is clearly less computationally-expensive, given that the time complexity of matrix inversion is typically $\mathcal{O}(n^3)$. This also indicates that if the same amount of computation is required for linear filtering, FCP can use a much longer filter (in time) than time-domain Wiener filtering.

## K  UNSSOR's effectiveness when used with other DNN architectures

We observe that the effectiveness of the proposed system comes from both the strong modelling capabilities of TF-GridNet and the contributions of the UNSSOR mechanism itself. Without using UNSSOR to deal with the ill-posed problem, the modelling capability of strong DNNs cannot be unleashed to separate speakers; and without using a strong DNN, the patterns in speech cannot be modelled well to realize good separation.

We expect UNSSOR to work with many DNN architectures, as long as the architecture is reasonably strong and can effectively deal with reverberation. To validate this, we experiment UNSSOR with DNN arachitectures with lower modelling capability. We select two representative separation models from the literature, TCN-DenseUNet [74] and DPRNN [75].

We replace TF-GridNet with the TCN-DenseUNet described in [74], and train the network using the same training configurations. TCN-DenseUNet [74] contains a temporal convolution network sandwiched by a UNet with DenseNet blocks. It is a reasonably strong separation model, which is fully convolutional and shares many similarities with many contemporary DNN architectures [79–83]. According to [23], it is worse than TF-GridNet in supervised separation tasks. We provide the unsupervised separation results in Table 9, which are obtained by using six-channel input and loss. We observe that UNSSOR works to some extent with TCN-DenseUNet, and the results are not as good as the ones in Table 1 obtained by using TF-GridNet.

The DPRNN architecture [75] in our experiments has a window size of 4 ms and a hop size of 1 ms. It has 6 layers. The number of bases is 256. The bottleneck dimension is 128. The number of hidden units in each BLSTM in each direction is 128. We apply ReLU as the encoder non-linearity and as the non-linearity for embedding masking. The chunk size is set to 64 and the chunk overlap is 50%. To leverage spatial information for model training, we follow the strategy proposed in [84] (see its Fig. 2 to get the idea), where spectral embeddings are learned together with spatial embeddings and

Table 9: Averaged results of 2-speaker separation on SMS-WSJ (6-channel input and loss), obtained by using **TCN-DenseUNet** [74] architecture.

| Row | Systems | $I$ | $J$ | Loss | Val. set SDR (dB) | Test set SDR (dB) | SI-SDR (dB) | PESQ | eSTOI |
|-----|---------|-----|-----|------|---------|---------|------------|------|-------|
| 0a | Mixture | - | - | - | 0.1 | 0.1 | 0.0 | 1.87 | 0.603 |
| 2a | UNSSOR | 19 | 0 | $\mathcal{L}_{MC+ISMS}$ | 11.1 | 10.7 | 9.7 | 2.92 | 0.770 |
| 2b | UNSSOR + Corr. based freq. align. | 19 | 0 | $\mathcal{L}_{MC+ISMS}$ | 11.1 | 10.7 | 9.7 | 2.92 | 0.770 |
| 2c | UNSSOR + Oracle freq. align. | 19 | 0 | $\mathcal{L}_{MC+ISMS}$ | 11.2 | 10.8 | 9.8 | 2.93 | 0.773 |
| 4a | PIT (supervised) | - | - | - | 14.6 | 13.9 | 13.4 | 3.58 | 0.878 |

Table 10: Averaged results of 2-speaker separation on SMS-WSJ (6-channel input and loss), obtained by using **DPRNN** [75] architecture.

| Row | Systems | $I$ | $J$ | Loss | Val. set SDR (dB) | Test set SDR (dB) | SI-SDR (dB) | PESQ | eSTOI |
|-----|---------|-----|-----|------|---------|---------|------------|------|-------|
| 0a | Mixture | - | - | - | 0.1 | 0.1 | 0.0 | 1.87 | 0.603 |
| 2a | UNSSOR | 19 | 0 | $\mathcal{L}_{MC+ISMS}$ | 9.2 | 8.9 | 8.0 | 2.68 | 0.724 |
| 2b | UNSSOR + Corr. based freq. align. | 19 | 0 | $\mathcal{L}_{MC+ISMS}$ | 9.2 | 8.9 | 8.0 | 2.68 | 0.724 |
| 2c | UNSSOR + Oracle freq. align. | 19 | 0 | $\mathcal{L}_{MC+ISMS}$ | 9.3 | 9.0 | 8.1 | 2.68 | 0.727 |
| 4a | PIT (supervised) | - | - | - | 12.3 | 11.7 | 11.3 | 3.00 | 0.820 |

DNN-estimated masks are used to mask spectral embeddings. In the six-channel case, the spatial embedding dimension is set to 360, following [84]. Differently from the Fig. 2 of [84], we do not use microphone-pair-wise Conv1D layers to obtain spatial embeddings. Instead, we obtain them by using a Conv1D layer with $P (= 6)$ input channels and 360 output channels. We first use the DPRNN to obtain intermediate separation results in the time domain, and then apply STFT (with a window size of 32 ms, a hop size of 8 ms, and the square-root of Hann window) to obtain $\hat{Z}(c)$ for each speaker $c$. The network is trained by only using the MC loss in (9) and not using the ISMS loss in (10). Although the ISMS loss is not used, we only observe minor frequency permutation on the SMS-WSJ dataset. All the other procedures are the same as the TF-GridNet based UNSSOR system. The results on SMS-WSJ, obtained by using six-channel input and loss, are presented in Table 10. We observe that UNSSOR also works with DPRNN to some degrees, and the results are not as good as the ones in Table 1.

## L   Comparison with MixIT

This section compares the results of UNSSOR with MixIT [30], which also deals with unsupervised separation. We put this comparison only in appendix, considering that MixIT may not be an ideal baseline to UNSSOR. Notice that MixIT needs to be trained on synthetic mixtures of mixtures (MoM), which ideally would require the two mixtures used in creating each MoM to be recorded in the same room using the same device. Differently, UNSSOR is designed to be trained directly on existing mixtures. In this case, the two models would be trained on different training examples, and this would make the comparison difficult. In the main body of this paper, we hence only consider methods that can be trained (or performed) directly on existing mixtures (such as IVA, spatial clustering and iRAS) as baselines.

To report the results of MixIT, we create a particular scenario (ideal for MixIT), where, for each existing SMS-WSJ mixture ($y = x(1) + x(2) + n$ with $n$ denoting noise), we randomly add two extra speakers in the same simulated room and use the same array placed at the same location so that we can have two 2-speaker mixtures (i.e., an existing SMS-WSJ mixture 1: $y^{(1)} = x^{(1)}(1) + x^{(1)}(2) + n^{(1)}$ and a newly-simulated mixture 2: $y^{(2)} = x^{(2)}(1) + x^{(2)}(2) + n^{(2)}$ where the superscript $(1)$ denotes mixture 1 and $(2)$ denotes mixture 2) to create an MoM for training MixIT for 4-speaker separation. Similarly to $n^{(1)}$, $n^{(2)}$ is sampled such that the SNR between $x^{(2)}(1) + x^{(2)}(2)$ and $n^{(2)}$ is in the range of $[20, 30]$ dB. The DNN architecture and training configurations for MixIT are the same as that in UNSSOR and PIT. The loss function is defined similarly to (5) on the real, imaginary and magnitude of the reconstructed mixtures weighted by the summation of mixture magnitudes. Similarly to our proposed technique, we can feed 1-, 3- or 6-channel mixtures to TF-GridNet-based MixIT. At run time, the trained MixIT model is used to separate the existing two-speaker mixtures in SMS-WSJ to four outputs, and the two outputs with the highest energy are selected for evaluation.

Table 11: Averaged results of 2-speaker separation on SMS-WSJ, obtained by using MixIT [30].

| Row | Systems | Val. set SDR (dB) | Test set SDR (dB) | SI-SDR (dB) | PESQ | eSTOI |
|-----|---------|---------|---------|---------|------|-------|
| 0a | Mixture | 0.1 | 0.1 | 0.0 | 1.87 | 0.603 |
| 5a | One-channel MixIT | 6.7 | 6.6 | 6.3 | 2.21 | 0.691 |
| 5b | Three-channel MixIT | 0.1 | 0.1 | 0.0 | 1.91 | 0.608 |
| 5c | Six-channel MixIT | 8.2 | 8.0 | 7.8 | 2.43 | 0.745 |

This way, the evaluation scores can be pretty much directly compared with the ones obtained by UNSSOR.

The results are shown in Table 11. They are not as good as the ones reported in Table 1, 2, 3 and 4. For unknown reasons, three-channel MixIT fits the loss well but fails at separating speakers.

