# OpenReview forum: "UNSSOR: Unsupervised Neural Speech Separation by Leveraging Over-determined Training Mixtures"
_NeurIPS.cc/2023/Conference — NeurIPS 2023 poster_

### Official Review · Reviewer_ux7k · 2023-06-12

**Soundness:** 3 good
**Presentation:** 2 fair
**Contribution:** 2 fair
**Rating:** 6
**Confidence:** 5

**Summary:**

This paper presents an innovative approach to unsupervised neural speech separation, leveraging the conditions where the number of microphones surpasses the number of speakers. The authors propose a method, named UNSSOR, which transforms an originally ill-posed problem - one that does not have a unique solution - into a well-posed problem - one with a unique solution - thereby facilitating the separation of speakers.

Key contributions of the paper include:

- The authors establish a linear-filter constraint between each speaker's reverberant images at each microphone pair, which converts the ill-posed problem into a well-posed one, thereby enhancing the separation of speakers.

- The authors devise loss functions inspired by the blind deconvolution problem and propose a DNN-based approach to optimize these functions. The speaker images are determined via DNNs, while the linear filters are estimated using a sub-band linear prediction algorithm named FCP, based on the mixture and DNN estimates.

- To address the frequency permutation issue that arises when using sub-band FCP, the authors propose a loss term that minimizes a measure known as intra-source magnitude scattering.

The authors claim that UNSSOR can be trained to perform under-determined separation, such as monaural unsupervised speech separation, based on over-determined training mixtures.

**Strengths:**

The authors propose a novel unsupervised neural speech separation method. The proposed method, UNSSOR, utilizes the multimicrophone over-determined condition to solve the unsupervised learning problem of speech separation. This is a completely new perspective compared to previous unsupervised speech separation methods. The authors have designed innovative loss functions that guide the unsupervised separation model about the desired sound objects and encourage the separation of speakers. This approach is original and shows a creative combination of existing ideas.

**Weaknesses:**

However, there are several limitations to this paper that undermine its true potential. Please see below for concerns and questions for the authors.

1. The article mentions that MixIT may have two mixtures that are not in the same scene under reverberant conditions, which in turn may result in providing spatial a priori information, or providing a priori information due to differences in the configuration and number of microphones. However, I feel that this statement is a bit too hypothetical. I think these problems are more a result of how the dataset is constructed and not a deficiency of the MixIT method itself. More specifically, these problems should be solved by constructing a room-specific dataset. The SMS-WSJ dataset mentioned in the paper can control the room's parameters, the arrangement of microphones, and the number of microphones. By finely designing and controlling these parameters, the resulting problems can be effectively circumvented, allowing the MixIT method to be evaluated in a more consistent environment. Therefore, I believe the problems mentioned in the paper do not serve as a weakness of the MixIT method.

2. In Section 4.2, I noticed that the authors proposed the MC loss function. However, as far as I understand, the MC loss function is not presented in this paper and has been described in detail in reference [1]. Cite [1] as a reference and clarify whether the MC loss function in the paper is an innovation from [1] and where exactly the innovation is.

3. The authors chose TF-GridNet as the separation DNN structure for the UNSSOR method.TF-GridNet has shown excellent performance in supervised separation tasks. However, the question I would like to raise is: Is the UNSSOR method's effectiveness due to the separation model TF-GridNet's strong performance, or is it due to the contribution of the UNSSOR method itself? I believe that to elaborate the capability of the UNSSOR method better; it is necessary to try to use different separation structures and compare their performance on each structure. If the UNSSOR method performs well on other separation structures, then we can be more confident that the UNSSOR method itself is highly robust and general. Such an analysis is necessary to assess and understand the actual value of the UNSSOR method.

4. I note that the approach in the paper combines traditional methods (Spatial clustering and IVA) with the DNN-based method iRAS. In addition, UNSSOR, a DNN-based scheme, is compared to include only one DNN method, which may not be comprehensive, in my opinion. Of particular note is that MixIT is a popular and widely used method for unsupervised speech separation, as shown in references [2-5]. I was puzzled while reading the paper as to why the authors did not include MixIT in the comparison.

5. The authors set different STFT window sizes and hop lengths in the comparison, which may raise questions about the fairness of the results. For the separation model of DNN, different settings of STFT window sizes and hop lengths can significantly affect the model's performance [6].

### References
[1]. Wisdom S, Hershey J R, Wilson K, et al. Differentiable consistency constraints for improved deep speech enhancement[C]//ICASSP 2019-2019 IEEE International Conference on Acoustics, Speech and Signal Processing (ICASSP). IEEE, 2019: 900-904.

[2]. Wisdom S, Tzinis E, Erdogan H, et al. Unsupervised sound separation using mixture invariant training[J]. Advances in Neural Information Processing Systems, 2020, 33: 3846-3857.

[3]. Tzinis E, Adi Y, Ithapu V K, et al. RemixIT: Continual self-training of speech enhancement models via bootstrapped remixing[J]. IEEE Journal of Selected Topics in Signal Processing, 2022, 16(6): 1329-1341.

[4] Tzinis E, Casebeer J, Wang Z, et al. Separate but together: Unsupervised federated learning for speech enhancement from non-iid data[C]//2021 IEEE Workshop on Applications of Signal Processing to Audio and Acoustics (WASPAA). IEEE, 2021: 46-50.

[5] Zhang J, Zorila C, Doddipatla R, et al. Teacher-student MixIT for unsupervised and semi-supervised speech separation[J]. arXiv preprint arXiv:2106.07843, 2021.

[6]  Peer T, Gerkmann T. Phase-aware deep speech enhancement: It's all about the frame length[J]. JASA Express Letters, 2022, 2(10): 104802.

**Questions:**

My detailed questions are as described above.

(1) Is the MC loss function in this paper an innovation from [1] and where is the innovation marked?

(2) Is the effectiveness of the UNSSOR method due to the powerful performance of the separation model TF-GridNet, or is it due to the contribution of the UNSSOR method itself?

(3) Why not compare MixIT method?

(4) Why not use the same STFT settings, especially compared to the DNN model iRAS.

**Limitations:**

The authors discuss the limitation of the proposed method.

---

> ### Author Rebuttal · Authors · 2023-08-09
>
> > (1) Is the MC loss function in this paper an innovation from [1] and where is the innovation marked?
>
> > [1]. Wisdom et al. Differentiable consistency constraints for improved deep speech enhancement, ICASSP, 2019.
>
> Sorry for the confusion. We think that it is a novel contribution, and we should have marked the innovations.
>
> We will emphasize in the paper that the proposed MC loss is very different from [1] in the following aspects.
>
> First, in [1], DNN estimates are strictly constrained to add up to the mixture (See Eq. (7) and (9) in [1]), but our MC loss only ``encourages'' the filtered DNN estimates to add up to the mixture.
>
> Second, our MC loss is applied to filtered DNN estimates rather than DNN estimates.
>
> Third, we are dealing with multi-microphone MC, while [1] only addresses single-channel.
>
> Overall, the proposed MC loss and the one in [1] have very different motivations and physical meanings.
>
> We now realize that it is not a good idea to use the same name, and we will change to ``mixture-constraint'' loss to indicate differences.
>
> > (2) Is the effectiveness of the UNSSOR method due to the powerful performance of the separation model TF-GridNet, or is it due to the contribution of the UNSSOR method itself?
>
> > I believe that to elaborate the capability of the UNSSOR method better; it is necessary to try to use different separation structures and compare their performance on each structure. If the UNSSOR method performs well on other separation structures, then we can be more confident that the UNSSOR method itself is highly robust and general. Such an analysis is necessary to assess and understand the actual value of the UNSSOR method.
>
> We think that both are important. Without using UNSSOR to deal with the ill-posed problem, the modelling capability of strong DNNs cannot be unleased to separate speakers; and without using a strong DNN, the patterns in speech cannot be modelled well to realize good separation.
>
> We think that the proposed UNSSOR method would not just work with a particular DNN architecture. We expect that the proposed UNSSOR mechanism can work with many DNN architectures, as long as the architecture is reasonably strong and can effectively handle reverberation, and that stronger DNN architectures would likely produce better separation.
>
> To further address the comments, we replace TF-GridNet with TCN-DenseUNet [A1], and train the network using the same training configurations. TCN-DenseUNet contains a temporal convolution network (TCN) sandwiched by a UNet with DenseNet blocks. It is a reasonably strong separation model, which is fully convolutional and shares many similarities with many modern DNN architectures in the literature, and, according to [A2], it is worse than the recent TF-GridNet architecture in supervised separation tasks. We provide the unsupervised results in Table 1 of the attached .pdf file. We observe that it obtains reasonably-good separation results in unsupervised tasks.
>
> [A1] Wang et al., Leveraging Low-Distortion Target Estimates for Improved Speech Enhancement, arXiv preprint arXiv:2110.00570, 2021.
>
> [A2] Wang et al., TF-GridNet: Integrating Full- and Sub-Band Modeling for Speech Separation, arXiv preprint arXiv:2211.12433, 2022.
>
> > (3) Why not compare MixIT method?
>
> > The article mentions that MixIT may have two mixtures that are not in the same scene under reverberant conditions, which in turn may result in providing spatial a priori information, or providing a priori information due to differences in the configuration and number of microphones. However, I feel that this statement is a bit too hypothetical. I think these problems are more a result of how the dataset is constructed and not a deficiency of the MixIT method itself. More specifically, these problems should be solved by constructing a room-specific dataset. Therefore, I believe the problems mentioned in the paper do not serve as a weakness of the MixIT method.
>
> > I note that the approach in the paper combines traditional methods with the DNN-based method iRAS. UNSSOR, a DNN-based scheme, is compared to include only one DNN method, which may not be comprehensive. Of particular note is that MixIT is a popular and widely used method for unsupervised speech separation, as shown in [2-5]. I was puzzled while reading the paper as to why the authors did not include MixIT in the comparison.
>
> See our response to all the reviewers.
>
> > (4) Why not use the same STFT settings, especially compared to the DNN model iRAS.
>
> > The authors set different STFT window sizes and hop lengths in the comparison, which may raise questions about the fairness of the results. For the separation model of DNN, different settings of STFT window sizes and hop lengths can significantly affect the model's performance [6].
>
> > [6] Peer et al. Phase-aware deep speech enhancement: It's all about the frame length, JASA, 2022.
>
> Just to clarify. For UNSSOR and iRAS, we use exactly the same STFT setting, i.e., 32 ms window size and 8 ms hop size.
> This STFT setting is very common in STFT-domain separation algorithms.
>
> If the reviewer meant different filter taps (in time) are used in UNSSOR and iRAS, Appendix G (see Fig. 4) is provided to address the concerns.
> The idea is that, since UNSSOR performs filtering in the time-frequency domain and iRAS performs filtering in the time domain, we configure the filter lengths (in time) to be the same for comparison.
>
> If the reviewer meant different STFT window and hop sizes are used for IVA and spatial clustering, we emphasize that it is very common for IVA and spatial clustering to use longer window so that enough amount of reverberation is covered in each frame. This way, their model assumption can be better satisfied and better separation can be achieved.
> It is very common in IVA and spatial clustering to tune STFT window and hop sizes.
> We will emphasize this in the paper.
>
> The referred paper [6] only considers non-reverberant cases, and therefore may not tell the full story.

---

> > ### Comment · Reviewer_ux7k · 2023-08-10
> > **Regarding Q2+Q3**
> >
> > Regarding Q2 - I think you should compare the speech separation approach rather than the speech enhancement approach. I would like to see the separation performance of classical speech separation models (e.g., Conv-TasNet or DPRNN) in the UNSSOR framework to assess the generality of UNSSOR better.
> >
> > Regarding Q3 - I look forward to your update on the results of MixIT's experiments to re-evaluate my recommended scores.

---

> > > ### Author Response · Authors · 2023-08-19
> > >
> > > > Regarding Q2 - I think you should compare the speech separation approach rather than the speech enhancement approach. I would like to see the separation performance of classical speech separation models (e.g., Conv-TasNet or DPRNN) in the UNSSOR framework to assess the generality of UNSSOR better.
> > >
> > > Thanks for the further comments.
> > >
> > > In the literature, TCN-DenseUNet has also been applied to speaker separation tasks and obtained reasonably strong performance. See for example [A1-A3] listed below.
> > >
> > > Following the criticism, we have also experimented UNSSOR with a six-channel DPRNN architecture.
> > >
> > > The DPRNN has a window size of $4$ ms and a hop size of $1$ ms.
> > > It has $6$ layers.
> > > The number of bases is $256$.
> > > The bottleneck dimension is $128$.
> > > The number of hidden units in each BLSTM in each direction is $128$.
> > > We apply ReLU as the encoder non-linearity and as the non-linearity for embedding masking.
> > > The chunk size is set to $64$ and the chunk overlap is $50$%.
> > > To leverage spatial information for model training, we follow the strategy proposed in [A4] listed below (see its Fig. 2 to get the idea), where spectral embeddings are learned together with spatial embeddings and DNN-estimated masks are used to mask spectral embeddings.
> > > In the six-channel case, the spatial embedding dimension is set to $360$, following [A4].
> > > Differently from Fig. 2 of [A4], we don't use microphone-pair-wise Conv1D layers to obtain spatial embeddings, and we obtain them by using a Conv1D layer with $P$ ($=6$) input channels and $360$ output channels.
> > > We first use the DPRNN to obtain intermediate separation results in the time domain, and then apply STFT (with a window size of $32$ ms, a hop size of $8$ ms, and the square-root of Hann window) to obtain $\hat{Z}(c)$ for each speaker $c$.
> > > The network is trained only using the MC loss in Eq. (9) of the paper and without using the ISMS loss in Eq. (10), while we only observe minor frequency permutation.
> > > All the other procedures are the same as the TF-GridNet based UNSSOR system.
> > >
> > > The results on SMS-WSJ (obtained by using six-channel input and loss) are shown below.
> > >
> > > Row | System | $I$ | $J$ | Loss | SDR (val.)| SDR | SI-SDR | PESQ | eSTOI
> > > | :----: | :----: | :----: | :----: | :----: | :-----: | :----: | :----: | :----: | :----: |
> > > 0a | Mixture | - | - | - | 0.1 | 0.1 | 0.0 | 1.87 | 0.603
> > > 2a | UNSSOR | 19 | 0 | $\mathcal{L}_{\text{MC}}$ | 8.7 | 8.7 | 7.8 | 2.64 | 0.719
> > > 2b | UNSSOR + Corr. based freq. align. | 19 | 0 | $\mathcal{L}_{\text{MC}}$ | 8.7 | 8.7 | 7.8 | 2.64 | 0.719
> > > 2c | UNSSOR + Oracle freq. align. | 19 | 0 | $\mathcal{L}_{\text{MC}}$ | 8.8 | 8.8 | 7.9 | 2.65 | 0.722
> > > 4a | PIT (supervised) | - | - | - | 12.3 | 11.7 | 11.3 | 3.00 | 0.820
> > >
> > > We observe that UNSSOR also works, to some extent, with DPRNN, in addition to TCN-DenseUNet. Both architectures have lower modelling capability than TF-GridNet.
> > >
> > > We will add these results to the paper.
> > >
> > > [A1] H. Taherian, K. Tan, and D. Wang, “Multi-Channel Talker-Independent Speaker Separation Through Location-Based Training,” IEEE/ACM TASLP, vol. 30, pp. 2791–2800, 2022.
> > >
> > > [A2] Z.-Q. Wang, G. Wichern, and J. Le Roux, "Convolutive Prediction for Monaural Speech Dereverberation and Noisy-Reverberant Speaker Separation", in IEEE/ACM TASLP, vol. 29, pp. 3476-3490, 2021.
> > >
> > > [A3] Y. Liu and D. Wang, “Divide and Conquer: A Deep CASA Approach to Talker-Independent Monaural Speaker Separation,” IEEE/ACM TASLP, vol. 27, no. 12, pp. 2092–2102, 2019.
> > >
> > > [A4] Zhang et al., “On End-to-End Multi-Channel Time Domain Speech Separation in Reverberant Environments,” in ICASSP, 2020, pp. 6389–6393.
> > >
> > > > Regarding Q3 - I look forward to your update on the results of MixIT's experiments to re-evaluate my recommended scores.
> > >
> > > We have now obtained the results of MixIT. See our responses to all the reviewers.

---

> > > > ### Comment · Reviewer_ux7k · 2023-08-19
> > > > **Final comments**
> > > >
> > > > Thanks to the authors for their in-depth experimentation with MixIT's versatility and exploration.
> > > >
> > > > I am raising my score from 5 to 6.

---

### Official Review · Reviewer_XWBZ · 2023-07-05

**Soundness:** 3 good
**Presentation:** 4 excellent
**Contribution:** 4 excellent
**Rating:** 7
**Confidence:** 4

**Summary:**

The most popular method for training neural networks for speech separation is by artificially mixing sources, since neural network based training requires supervision. However, this kind of supervised training creates a mismatch since the mixtures seen at test time contain real overlapping speech and noise. This paper formalizes the problem of unsupervised speech separation, and posits that it becomes a well-posed problem for over-determined conditions (i.e., when the number of microphones is more than the number of speakers). The authors propose a method (inspired from signal processing) that uses the images at each microphone as supervision for the neural network training, and a new loss function to address the frequency permutation problem that often happens in blind source separation (BSS). They also propose a method to use these models for monoaural separation.

**Strengths:**

Overall, I believe this is an extremely strong paper with landmark results for unsupervised speech separation. As mentioned above, the majority of literature in the field of speech separation, since permutation-invariant training (PIT) was proposed, has only revolved around better neural network architectures. While the SI-SDR on the synthetic WSJ-Mix benchmark has consequently gone up, significantly less attention has been paid to real overlapping speech. The paradigm of continuous speech separation (CSS) tries to address some of the issues (such as sparse overlaps), but it is also trained with PIT on synthetic mixtures. The result is that there exists no public speech separation models that can perform well on multi-speaker data such as AMI, CHiME-6, AliMeeting, etc. I think UNSSOR is quite promising and should change the face of speech separation research.

The paper is strong on many levels, such as the following.

1. The authors correctly identify that the problem with training separation networks has to do with a lack of accurate supervision, and propose that this supervision may come from estimating each speaker’s reverberant image at every microphone. Under the assumption of a linear-filter constraint between a speaker’s images at the microphones, the resulting linear system has a unique solution, as described in Section 3.

2. The authors design loss functions which are inspired from signal processing formulation of the problem. This includes (i) mixture consistency loss for the filtered estimates, (ii) forward convolutive prediction (FCP) for relative RIR estimation, (iii) causal/non-causal stacking to handle time misalignment, and (iv) an intra-source magnitude scattering loss to address the frequency permutation problem. Overall, all these losses have clear motivations and design.

3. The recently proposed TF-GridNet encoder is used as the backbone of the model. This architecture has been shown to obtain state-of-the-art results on PIT-based separation (even surpassing time-domain models), and it is interesting to see that it can be trained well with the unsupervised technique.

4. The authors compare their method with extremely strong baselines, which makes their conclusions stronger. For example, they implemented a novel variant of the recently proposed “Reverberation as Supervision” (RAS) method, called iRAS, which can perform unsupervised separation. This modification could very well be a short paper in itself.

5. The results on SMS-WSJ show that UNSSOR obtains results approaching supervised separation models. For example, for the 6-channel setting, the SDR on the test set is 15.6 dB (compared to 19.4 dB for PIT model). As a comparison, the next unsupervised method is IVA, which obtains 10.6 dB. Moreover, on reducing the number of channels from 6 to 3, UNSSOR’s performance degrades only marginally (to 15.4 dB), whereas PIT degrades to 16.8 dB.

6. The authors even find a way to use models trained with multi-channel inputs for performing mono-aural separation.

7. The code and models for UNSSOR will be released.

A side effect of all of the above is that the paper is quite dense and may not be accessible to readers who do not have a source separation background. However, it is a case study in how neural methods can be designed by incorporating the knowledge from other domains (in this case, signal processing).


**Weaknesses:**

1. The main point of concern about the paper is that results are shown just for the SMS-WSJ dataset. Such an evaluation defeats the original motivation for unsupervised training, since SMS-WSJ contains synthetically mixed sources. Clearly, the PIT models outperform UNSSOR on this benchmark, which is expected. The real test of UNSSOR would have been an evaluation on real mixtures, such as AMI, CHiME-6, or AliMeeting, where PIT-based models fail completely. Nevertheless, since the paper proposes a completely novel paradigm of speech separation, I am willing to forgo this point, but at the cost of docking a point from the ratings.

2. Can UNSSOR be trained with data containing different numbers of channels or different array geometries? It seems that the choice of the hyper-parameters $I$ and $J$ may be heavily dependent on the microphone configuration.

3. The authors delegate the “Limitations” section to the appendix. I think it should be put in the main paper, since an important part of presenting our research is talking about its limitations. This would also be useful for other researchers to get ideas for building upon UNSSOR for their own work.


**Questions:**

1. How robust is UNSSOR to change in array configuration or the number of channels for the case of mono-aural separation?

2. In Section 5.2, can you briefly explain what these metrics are, for the unfamiliar reader.


**Limitations:**

None, except those presented in Appendix H.

---

> ### Author Rebuttal · Authors · 2023-08-09
>
> > Overall, I believe this is an extremely strong paper with landmark results for unsupervised speech separation. As mentioned above, the majority of literature in the field of speech separation, since permutation-invariant training (PIT) was proposed, has only revolved around better neural network architectures. While the SI-SDR on the synthetic WSJ-Mix benchmark has consequently gone up, significantly less attention has been paid to real overlapping speech. The paradigm of continuous speech separation (CSS) tries to address some of the issues (such as sparse overlaps), but it is also trained with PIT on synthetic mixtures. The result is that there exists no public speech separation models that can perform well on multi-speaker data such as AMI, CHiME-6, AliMeeting, etc. I think UNSSOR is quite promising and should change the face of speech separation research.
>
> > The paper is strong on many levels, such as the following.
>
> > ...
>
> > A side effect of all of the above is that the paper is quite dense and may not be accessible to readers who do not have a source separation background. However, it is a case study in how neural methods can be designed by incorporating the knowledge from other domains (in this case, signal processing).
>
> We feel so excited after reading the comments!
>
> This is such a big encouragement for us to invest more on improving the proposed methods in follow-up studies.
>
> > The main point of concern about the paper is that results are shown just for the SMS-WSJ dataset. Such an evaluation defeats the original motivation for unsupervised training, since SMS-WSJ contains synthetically mixed sources. Clearly, the PIT models outperform UNSSOR on this benchmark, which is expected. The real test of UNSSOR would have been an evaluation on real mixtures, such as AMI, CHiME-6, or AliMeeting, where PIT-based models fail completely. Nevertheless, since the paper proposes a completely novel paradigm of speech separation, I am willing to forgo this point, but at the cost of docking a point from the ratings.
>
> Insightful point!
>
> We made an effort to evaluate on AMI, CHiME-6, and AliMeeting datasets, which are real-recorded meeting-style data. However, to obtain strong performance, we need to deal with many other problems (such as sparse speaker overlap, varying number of speaker, unknown number of speakers etc.), which would make the paper much less focused. We hence leave the evaluation as a future work, and this paper focuses on showing the potential of UNSSOR, which will be the core technique we will build upon in our future work.
>
> > Can UNSSOR be trained with data containing different numbers of channels or different array geometries? It seems that the choice of the hyper-parameters and may be heavily dependent on the microphone configuration.
>
> Good point!
>
> We think that UNSSOR can be trained with the mentioned data, and this could be a good future extension.
>
> We could use DNNs that can handle variable number of input channels and different array geometries.
>
> The loss can also be computed on training examples that have different numbers of channels, by configuring the filter taps in a smart way to deal with different array geometries (i.e., to cover a wide range of microphone distances).
>
> > The authors delegate the “Limitations” section to the appendix. I think it should be put in the main paper, since an important part of presenting our research is talking about its limitations. This would also be useful for other researchers to get ideas for building upon UNSSOR for their own work.
>
> Will change.
>
> > How robust is UNSSOR to change in array configuration or the number of channels for the case of mono-aural separation?
>
> At run time, the trained model performs monaural separation. We think that it would be invariant to changes in array configurations or the number of channels.
>
> At training time, we expect that UNSSOR can robustly deal with training examples with various array configurations or number of channels.
> Notice that the DNN is trained to only exploit monaural spectro-temporal patterns for separation. As long as there is a sufficient number of microphone mixtures to help pinpoint the solutions to speaker images, we expect the DNN to be trained well, although the shape of the loss surface afforded by different array configurations or numbers of channels could influence the training. This investigation could be a follow-up paper.
>
> We expect that using more microphones for loss computation would lead to clearly better separation, but we don't observe this in our current evaluations (e.g., see Table 3 and 4). We will investigate this in a future study.
>
> > In Section 5.2, can you briefly explain what these metrics are, for the unfamiliar reader.
>
> They are popular metrics in speech separation. SI-SDR and SDR measure the quality of predictions at the sample level, and PESQ and eSTOI are objective metrics of speech quality and intelligibility respectively.
>
> We will add this description to the paper.

---

> > ### Comment · Reviewer_XWBZ · 2023-08-10
> > **Response to rebuttal**
> >
> > I thank the authors for clarifying some of my original points. I don't think this needs much more discussion, and I congratulate the authors on a remarkable paper!

---

### Official Review · Reviewer_XDQW · 2023-07-06

**Soundness:** 4 excellent
**Presentation:** 4 excellent
**Contribution:** 3 good
**Rating:** 6
**Confidence:** 5

**Summary:**

This paper proposes an unsupervised method for training neural networks to separate speech from multi-microphone recordings. The idea is to separate a reference microphone into separate sources, then use each microphone as a mixture signal that must match filtered versions of the reference microphone separated sources. The linear filters are computed in subbands, which results in a well-known frequency permutation problem, for which an additional "intra-source magnitude" loss term is proposed to ameliorate. Though the method requires "overdetermined" mixtures (i.e. more mics than sources), the method can also be used for underdetermined cases, e.g. 2 speakers with 1 mic, where a single mic is used as input, and multiple mics are used in the loss function. For 3-mic and 6-mic mixtures, the proposed method is shown to achieve better performance in terms of SDR, SI-SDR, PESQ, and eSTOI versus a baseline (the RAS algorithm) that works on a similar principle on the SMS-WSJ dataset. The method does not outperform a supervised PIT method, which is expected.

**Strengths:**

S1) The method is intuitive and is explained well, and the paper provides thorough description of the method and chosen hyperparamters.

S2) The proposed method can perform unsupervised training on single mixtures, unlike MixIT which requires combining at least two mixtures into a training examples.

S3) The authors clearly describe the differences of the proposed UNSSOR method with the RAS method, particularly why UNSSOR can successfully train unsupervised, while RAS cannot.

**Weaknesses:**

W1) The dataset used, SMS-WSJ, consists entirely of synthetic mixtures that use synthetic simulated RIRs. Thus, there is some concern that the method may not generalize to real-world acoustic scenarios. Some evaluation on real data would further improve the paper. Also, one of the main advantages of unsupervised algorithms is to be able to adapt separation models to unsupervised real-world data, so that models can work better on those data domains. Of course, evaluation on real domains cannot be evaluated by intrusive metrics, as used in this paper, and require subjective human listening tests. I am glad the authors are considering this as a future direction, and I encourage them towards that goal.

W2) As acknowledged by the authors, the method assumes stationary sources, which is potentially quite limiting when considering real scenarios where speakers may be moving. Even if speakers are seated, head movement can still produce acoustic effects. There are likely interesting extensions of the method to handle nonstationary spatial scenarios, such as allowing for slowly-varying linear filters.

W3) No audio demos are provided. Providing such demos would make it a lot easier to readers to evaluate the quality of the predictions and improve understanding of the method.

W4) It would be interesting to see how a single-channel unsupervised method like MixIT compares to the multi-microphone method. Also, since this paper was submitted, a multichannel version of MixIT seems to have been proposed (https://arxiv.org/abs/2305.11151), which seems to operate differently (network with multichannel input and multichannel output, and MixIT applied directly to multichannel outputs using multi-channel mixtures-of-mixtures). It would be interesting to compare both single-channel and multi-channel MixIT to the proposed approach; these MixIT methods will likely be mismatched to test time since they train on mixtures-of-mixtures, but it would be interesting to compare.

Minor comments and typos

a) "boot-start" -> "bootstrap" ?

b) "comptue" -> "compute"

c) Would be good to mention what \mathcal{F} is in description of equation (4)

d) "doing this would complicates" -> "doing this would complicate"

e) "assumed time-variant." -> "assumed time-variance."

f) Maybe bold the best numbers in each column for Tables 1 and 2?

**Questions:**

Q1) How much tuning of the number of filter taps was done? This seems like it could have a big effect on the performance of the model. Also, does tuning the number of taps suggest some knowledge about the unsupervised data? I suppose a practitioner could use a blind T60 model to estimate the distribution of T60 across an unsupervised dataset, which could give some insight into optimal filter taps. Besides T60 (i.e. duration of expected RIRs to model), are there other issues that affect the optimal choice of filter taps?

Q2) How certain is it that frequency permutation is the primary cause of the drop in performance going from rows 1a->1b and 2a->2b in Tables 1 and 2? The appendix provides an illustrative example, but none of the objective metrics are directly measuring frequency permutation. It seems that an objective metric could be formulated that directly measures the degree of frequency permutation, but perhaps such a metric is not necessary if spot checks of the predictions indicated that frequency permutation was present?

**Limitations:**

Limitations are discussed well.

---

> ### Author Rebuttal · Authors · 2023-08-09
>
> > W1) The dataset used, SMS-WSJ, consists entirely of synthetic mixtures that use synthetic simulated RIRs. Thus, there is some concern that the method may not generalize to real-world acoustic scenarios. Some evaluation on real data would further improve the paper.
> Also, one of the main advantages of unsupervised algorithms is to be able to adapt separation models to unsupervised real-world data, so that models can work better on those data domains. Of course, evaluation on real domains cannot be evaluated by intrusive metrics, as used in this paper, and require subjective human listening tests. I am glad the authors are considering this as a future direction, and I encourage them towards that goal.
>
> Great comment!
>
> We are currently working towards that goal, and will share our findings.
>
> > W2) As acknowledged by the authors, the method assumes stationary sources, which is potentially quite limiting when considering real scenarios where speakers may be moving. Even if speakers are seated, head movement can still produce acoustic effects. There are likely interesting extensions of the method to handle nonstationary spatial scenarios, such as allowing for slowly-varying linear filters.
>
> Yes, we will investigate the moving-source case, which is a common problem also in spatial processing.
>
> > W3) No audio demos are provided. Providing such demos would make it a lot easier to readers to evaluate the quality of the predictions and improve understanding of the method.
>
> An audio demo is available at https://anonymauth.github.io/
>
> We will add this link to the paper.
>
> > W4) It would be interesting to see how a single-channel unsupervised method like MixIT compares to the multi-microphone method. Also, since this paper was submitted, a multichannel version of MixIT seems to have been proposed (\url{https://arxiv.org/abs/2305.11151}), which seems to operate differently (network with multichannel input and multichannel output, and MixIT applied directly to multichannel outputs using multi-channel mixtures-of-mixtures). It would be interesting to compare both single-channel and multi-channel MixIT to the proposed approach; these MixIT methods will likely be mismatched to test time since they train on mixtures-of-mixtures, but it would be interesting to compare.
>
> See our response to all the reviewers.
>
> Thanks for pointing us the paper on multi-channel PIT. We plan to compare with multi-channel MixIT in a future study, as it was online after the submission of this paper.
>
> > a) "boot-start" -> "bootstrap" ?
>
> > b) "comptue" -> "compute"
>
> > c) Would be good to mention what $\mathcal{F}$ is in description of equation (4)
>
> > d) "doing this would complicates" -> "doing this would complicate"
>
> > e) "assumed time-variant." -> "assumed time-variance."
>
> > f) Maybe bold the best numbers in each column for Tables 1 and 2?
>
> Will change following the comments.
>
> > Q1) How much tuning of the number of filter taps was done?
> > This seems like it could have a big effect on the performance of the model.
> > Also, does tuning the number of taps suggest some knowledge about the unsupervised data?
> > I suppose a practitioner could use a blind T60 model to estimate the distribution of T60 across an unsupervised dataset, which could give some insight into optimal filter taps.
> > Besides T60 (i.e. duration of expected RIRs to model), are there other issues that affect the optimal choice of filter taps?
>
> Great question!
>
> The filter taps in Fig. 4 of the Appendix show the filter taps we tuned. It indeed has a big effect.
>
> Our experience is that the filter tap can not be set too long or too short. If it is set too long, the model would optimize the loss well but not separate speakers (i.e., overfit); and if it is set too short, the model would not fit the loss well (i.e., underfit).
>
> We see what you mean by using a blind T60 model to estimate the filter length for each training example, but the FCP filters in the study are the relative RIRs among speaker images at closely-placed microphones rather than the RIRs between sound source and its far-field images. We currently do not have a good way to determine the optimal choice of filter taps.
>
> > Q2) How certain is it that frequency permutation is the primary cause of the drop in performance going from rows 1a->1b and 2a->2b in Tables 1 and 2?
> > The appendix provides an illustrative example, but none of the objective metrics are directly measuring frequency permutation.
> > It seems that an objective metric could be formulated that directly measures the degree of frequency permutation, but perhaps such a metric is not necessary if spot checks of the predictions indicated that frequency permutation was present?
>
> We are confused by the first question. In Table 1 and 2, going from 1a to 1b and going from 2a to 2b both do not have performance drop.
>
> A direct metric of frequency permutation may not be necessary, as you suggested. Our spot checks are that frequency permutation is always presented, and its presence also makes sense as FCP is performed independently in each frequency. From the results in row 1a and 1c of Table 1 and 2, we can see that using oracle frequency permutation produces large improvement. This largely indicates how severe the frequency permutation problem is.

---

### Official Review · Reviewer_jddy · 2023-07-06

**Soundness:** 3 good
**Presentation:** 4 excellent
**Contribution:** 3 good
**Rating:** 7
**Confidence:** 4

**Summary:**

In this paper the authors tackle speech source separation, in which multiple fixed speech sources $X(c)$ are recorded by an array of p microphones, resulting in p observable mixtures $Y_p$. The authors propose an STFT domain source separation algorithm called UNSSOR, leveraging a complex-valued deep neural network (TF-GridNet) to estimate the individual sources at a reference microphone (e.g., mic 1). Typically the problem is solved in a supervised fashion using PIT (permutation invariant training) where the clean sources must be known, but the authors propose an unsupervised method, where only the mixtures $Y_p$ are required at training time. The key idea is to relax the under-determined problem, by leveraging physical constraints, i.e. the sources combining at a microphone p are the same sources at the reference microphone convolved with a relative room impulse response (relative RIR). As such the authors estimate virtual sources $Z(c)$ and relative RIRs at each microphone for each source $g_p(c)$. The relative RIRs are computed in closed form using FCP (a least squares optimization problem), leveraging only the mixtures $Y_p$ and the estimates $Z(c)$. Multiplying $g_p(c)$ with $Z(c)$ the authors obtain the FCP-estimated source images, which are summed to obtain mixtures estimates $\hat Y_{p}$ at each microphone, obtaining a mixture consistency loss $L_{MC}$ for training the neural network. The authors notice that solutions obtained with the model trained on $L_{MC}$ only exhibit the frequency permutation problem, where estimates on certain spectral bands are swapped between estimates. To address this problem, the authors regularize the variance of the magnitude of the FCP-estimated source images over frequency bands during training time, with a novel intra-source magnitude scattering loss $L_{ISMS}$. Combining $L_{MC}$ and $L_{ISMS}$ results in improved separation metrics. Finally, the authors try the algorithm in the monoaural unsupervised source separation setting, where they provide only one mixture at training time, but optimize with multiple microphones in the loss.

**Strengths:**

- The idea of reducing the number of equations using physical constraints is natural, leading to a well-structured unsupervised multi-mixture speech source separation algorithm. Computing the RIRs with least squares also is very useful, given that in such a way the mixture consistency loss depends only on the observed mixtures and the estimated virtual sources. The authors present such an idea in a very clear and formal manner, with relevant links to the existing literature. I believe that this approach is way more founded theoretically than the MixIt approach, leading to a better methodological line of research on unsupervised neural source separation.
- The intra-source magnitude scattering regularizer ($L_{ISMS}$) seems very interesting because such a loss can be incorporated not only in the presented multi-microphone speech scenario but in every source separation task working on STFT spectrograms (e.g, music, universal). Improving the results for more than 4dB gives strong empirical results to the effectiveness of such a technique.
- The method achieves good empirical results with respect to a plethora of metrics (SDR, SI-SDR, PESQ, eSTOI) on SMS-WSJ compared to other unsupervised multi-mic algorithms (Spatial clustering, IVA, RAS) and fares well with respect to a supervised PIT baseline (using the same neural architecture).

**Weaknesses:**

- UNSSOR, while improving the results in the monoaural speech separation setting with respect to iRAS (improves 1.4 dB on the SMS-WSJ test set both with 3 and 6 channels loss), still faces a large gap with respect to the supervised Monaural PIT baseline (~ 4 dB). This is expected since not relying on supervision can negatively impact the separation results. The authors should have included at least a comparison with some other popular unsupervised source separation baseline such as MixIt, in order to prove the superiority of UNSSOR in the unsupervised source separation arena.
- As with other unsupervised neural source separators such as MixIt, it is not really clear why one should go unsupervised if the supervised metrics are already better on the same dataset. These types of studies (including MixIt or its regularized versions such as https://arxiv.org/abs/2106.00847), should experimentally showcase that performing unsupervised source separation can benefit over supervised source separation as the dataset size increase. For example in https://arxiv.org/abs/2106.00847, authors cannot beat a supervised baseline trained on FUSS, using an order of magnitude more data (Audioset or YFCC100m) with an unsupervised method (regularized MixIt). Providing a scaling law should be of uttermost importance in papers such as the presented one and in future papers, if not we can continue inventing unsupervised methods that will never surpass supervised or weakly supervised methods (using labels, learning Bayesian priors).

**Questions:**

I will add suggestions for the improvement of the paper and ask relevant questions.

- Line 65: The authors should use a term like `metric` instead of `measure`, given the precise meaning of such a term in measure-theory. I know they can be synonyms in an applied field such as audio processing but it is better to be more precise.
- Line 120: `Hermitian transpose` instead of `Hermittan`
- Line 146: the term mixture consistency is defined here: https://arxiv.org/pdf/1811.08521.pdf, I think it's ok to use the same term but could create a little bit of confusion.
- Line 161: Can the authors explain briefly in the text, for better understanding, the rationale of matching the FCP-estimated images with the mixture in Eq. (6)?
- Line 167: Regarding the $\xi$ hyperparameter, I don't understand if it multiplies only the max or all the expression
- Line 179: I found the causal analysis well developed, but I still do not well understand why if a source sound c reaches the reference microphone earlier it should be processed with future values (non-causal filtering).
- Line 193: Given that $Z(c)$ is learned, why it is interpreted like a virtual microphone estimate and not the real dry signal at speaker $c$?
- Line 232: In the monoaural speech source separation setting, I don't understand what you match on the different microphones in the mixture consistency loss, when using only one input $Y_p=1$? Do you match the only available $Y_p=1$ at all "virtual" microphones?

**Limitations:**

The authors address the limitations of their work in Appendix H, namely that they assume to know the number of sources, they assume they are directional point sources at fixed positions. I do not believe these limitations are much problematic as the main limitation is the unavailability of data required for scaling up the algorithm with respect to the supervised baseline (if it scales).

---

> ### Author Rebuttal · Authors · 2023-08-09
>
> > The authors should have included at least a comparison with some other popular unsupervised source separation baseline such as MixIt
>
> See our response to all the reviewers.
>
> > As with other unsupervised neural source separators such as MixIt, it is not really clear why one should go unsupervised if the supervised metrics are already better on the same dataset. These types of studies (including MixIt or regularized MixIt https://arxiv.org/abs/2106.00847 [Wisdom'21]) should experimentally show that performing unsupervised separation can benefit over supervised separation as the dataset size increase. For example in [Wisdom'21], authors cannot beat a supervised baseline trained on FUSS, using an order of magnitude more data with an unsupervised method.
> > Providing a scaling law should be of uttermost importance in papers such as the presented one and in future papers, if not we can continue inventing unsupervised methods that will never surpass supervised or weakly supervised methods.
>
> Insightful comment! We will pay more attention to the scaling law in follow-up studies.
>
> In speech separation, a major motivation of using unsupervised separation is that the models can be trained directly on real mixtures (where the clean speech signals are not available) and hence could have better generalizability on real data than supervised models, which need to be trained on simulated data (often mismatched with real-recorded test data).
> As is also pointed out by reviewer ``XWBZ'', supervised models such as PIT have limited success so far on real-recorded multi-speaker datasets such as AMI, CHiME-6 and AliMeeting, even if a lot of data can be simulated to train PIT.
> A possible initial step towards solving this problem could be using an algorithm like UNSSOR which can be trained directly on real mixture and avoid using unrealistic synthetic mixtures for training.
>
> On the other hand, unsupervised separation and supervised separation may not be mutually exclusive, and they could be combined. We could, for example, adapt a supervised model to new domains via unsupervised mechanisms such as UNSSOR, or fine-tune unsupervised models such as UNSSOR via supervised mechanisms like PIT in a target domain, where some high-quality labelled data is available.
>
> > Line 65: Ue metric instead of measure
> > Line 120: Hermitian transpose instead of Hermittan
>
> Will change.
>
> > Line 146: the term mixture consistency is defined in [Wisdom 2019], I think it ok to use the same term but could create confusion.
>
> Will rename the loss to "mixture-constraint" loss to avoid any confusion.
>
> > Line 161: Can the authors explain briefly the rationale of matching the FCP-estimated images with the mixture in Eq. (6)?
>
> Will explain.
>
> If $Y_p$ only contains $X_p(c)$, (6) can estimate the relative RIR relating $\hat{Z}(c)$ to $X_p(c)$.
>
> If $Y_p$ contain other sources besides $X_p(c)$, (6) could still estimate the relative RIR following the derivations in Appendix C.
>
> > Line 167: Regarding the $\xi$, I don't understand if it multiplies only the max or all the expression
>
> Only to the max expression. We will make the equation clearer.
>
> > Line 179: I found the causal analysis well developed, but I still do not well understand why if a source sound c reaches the reference mic earlier it should be processed with future values (non-causal filtering).
>
> We give an example below.
>
> In Fig. 1(a) (see attached .pdf file), suppose that the blue signal is the DNN estimate for speaker $c$, and the orange signal is speaker $c$'s image at another microphone, which is a delayed version (i.e., reaching the microphone later). To filter the blue signal to approximate the oracle signal, we only need a causal filter.
>
> Reversely, suppose that the orange signal is the DNN estimate for speaker $c$, and the blue signal is speaker $c$'s image at another microphone, which is an advanced version (i.e., reaching the microphone earlier). To filter the orange signal to approximate the blue signal, we need a non-causal filter.
>
> We will add this example to the Appendix of the paper.
>
> > Line 193: Given that $Z(c)$ is learned, why it is interpreted like a virtual microphone estimate and not the real dry signal at speaker $c$?
>
> In Eq. (9), $\hat{Z}(c)$ is constrained such that it can be filtered by a causal filter $\hat{\mathbf{g}}_p(c)$ to approximate $X_p(c)$, and it is not explicitly constrained to be a dry signal. Since there could be an infinite number of $\hat{Z}(c)$ and $\hat{\mathbf{g}}_p(c)$ whose convolution results would well approximate $X_p(c)$, $\hat{Z}(c)$ is likely not the dry source signal.
>
> See Fig. 1(b)  (see attached .pdf file) for an example, where each virtual microphone captures the direct-path signal of a target speaker earlier than any other microphones so that we can use causal FCP filters.
>
> > Line 232: In the monoaural speech source separation setting, I don't understand what you match on the different microphones in the mixture consistency loss, when using only one input $Y_{p=1}$? Do you match the only available $Y_{p=1}$ at all "virtual" microphones?
>
> We now think that, in the monaural case, $\hat{Z}(c)$ would all be aligned to the speakers' images at the reference microphone $1$, since the DNN only has monaural input and in this case the DNN is not likely to align its outputs to a virtual microphone different from microphone $1$.
>
> To help understanding, we give an example in Fig. 1(c) (see attached .pdf file), where the reference microphone captures speaker $2$'s direct-path signal later than all the other microphones. In this case, we need to use non-causal FCP filters when filtering $\hat{Z}(c)$ (which is estimated based on the monaural signal at the reference microphone) to approximate speaker $2$'s images captured at the other microphones.
>
> > ... the main limitation is the unavailability of data required for scaling up the algorithm with respect to the supervised baseline (if it scales).
>
> See our responses to your earlier comment on this.

---

### Official Review · Reviewer_2uqn · 2023-07-06

**Soundness:** 3 good
**Presentation:** 3 good
**Contribution:** 3 good
**Rating:** 7
**Confidence:** 5

**Summary:**

The focus of this paper is unsupervised speech separation by exploiting training mixtures with more microphones than speakers (over-determined). The proposed neural separator is using the input mixture to constrain the estimated images of each speaker. It is proposed to also train the system for under-determined scenarios, e.g., for single-channel speech separation, by separating a single-channel signal, while using multichannel loss function.

**Strengths:**

The idea of using multichannel mixtures for unsupervised training is novel and reasonably effective compared to the supervised and unsupervised baselines.
The experimental evaluation is sufficient and demonstrates the effectiveness and includes sufficient data to support some of the design choices.
The paper is well written.

**Weaknesses:**

It would be useful for the readers less familiar with speech processing to include references to a few papers very relevant to relative IR estimation, mixture consistency and linear prediction.

**Questions:**

Only measurement/modeling noise is included in the model, and background noise or discrete noise sources would complicate things, e.g., since the same RTF could not be applied to the noise component. Is there any initial data on robustness in presence of noise?

ll. 119
It is claimed that the relative room impulse response in (2) is typically short. However, it should be noted that this is not necessarily true, even for the microphone array and reverberation times used in the experiments (20 cm diameter, up to 0.5s RT60). A useful datapoint is paper by [Talmon, 2009], Fig. 7 in particular. I think this it would be helpful for many readers at NeurIPS to mention this (or a similar paper) and point out that this assumption does not always hold, and has been investigated in the literature.

[Talmon, 2009] Talmot, Cohen, Gannot, Relative Transfer Function Identification Using Convolutive Transfer Function Approximation, IEEE Tr. ASLP, 2009.

ll. 128
- Typo in “comptue”

Section 4.2
Mixture consistency has already been proposed in [Wisdom, 2019]. This paper should be clearly mentioned. Also, consider renaming to multichannel mixture consistency, since you’re using P channels and RTFs.

[Wisdom, 2019] Wisdom et al., Differentiable Consistency Constraints for Improved Deep Speech Enhancement, 2019.

Section 4.3
Formulation is (6) is basically multichannel linear prediction with using estimated sources Z instead of the original mixtures Y. Relevant works solving a problem analogous to (6) should be at least briefly mentioned, such as earlier work in [Yoshioka, 2010] and many later works.

[Yoshioka, 2010] Yoshioka et al., Blind Separation and Dereverberation of Speech Mixtures by Joint Optimization, IEEE Tr. ASLP 2011.

**Limitations:**

The main limitation is that the model does not take into account background or discrete noise, and is only demonstrated to work in presence of uncorrelated (measurement) noise. This severely limits applicability to using real-world recordings to train a separation model.
It would be very useful to include results on robustness in presence of higher levels of noise. However, this is not essential and it may be out of scope of this paper.
Another limitation for real-world use is the assumption of a static scenario (fixed relative IRs), which would not hold in real recordings.

---

> ### Author Rebuttal · Authors · 2023-08-09
>
> > It would be useful for the readers less familiar with speech processing to include references to a few papers very relevant to relative IR estimation, mixture consistency and linear prediction.
>
>  Will include, especially [Talmon, 2019].
>
> > Only measurement/modeling noise is included in the model, and background noise or discrete noise sources would complicate things, e.g., since the same RTF could not be applied to the noise component. Is there any initial data on robustness in presence of noise?
>
> Yes, we need to consider background noises. We haven't had comprehensive results on this yet, and plan to address this in a follow-up study.
>
> We could use a number of garbage sources to absorb directional sources.
> That is, we can train UNSSOR to separate the mixture to $C+N$ sources (where $C$ is the hypothesized number of speakers and $N$ the hypothesized number of directional noise sources) so that we can have a separate RTF for each source.
> We envision that our method could be effective at separating a large number of directional sources (including both speech and noise sources) if there is a sufficient number of microphones to afford over-determined conditions.
> In this case, we can assume $\varepsilon$ to be a measurement/modelling noise, as the speech and noise signals are all modelled by the $N+C$ sources.
>
> > ll. 119
> > It is claimed that the relative room impulse response in (2) is typically short. However, it should be noted that this is not necessarily true, even for the microphone array and reverberation times used in the experiments (20 cm diameter, up to 0.5s RT60). A useful datapoint is paper by [Talmon, 2009], Fig. 7 in particular. I think this it would be helpful for many readers at NeurIPS to mention this (or a similar paper) and point out that this assumption does not always hold, and has been investigated in the literature.
>
> > [Talmon, 2009] Talmot et al., Relative Transfer Function Identification Using Convolutive Transfer Function Approximation, IEEE TASLP, 2009.
>
> Thanks for pointing this out. We realize that our sentence "Note that $\mathbf{g}_p(c,f)$ is very short (i.e., $E$ is small)" may be misleading.
>
> Just to clarify. We intended to say that $E$ is small. In our paper, given that the STFT hop size is $8$ ms, $E=I+1+J$ equals $20$ in in Table 1-2, and equals $21$ in Table 3-4. We don't mean that the RIR in the time domain is very short.
>
> Thanks for sharing the paper by Talmon et al. Based on our understanding of the paper, its Fig. 7(a) suggests that, in low-noise cases, when the microphone distance is larger, CTF, which can use multiple taps, can better estimate the RTF than MTF, which is restricted to only use one tap. In [Talmon, 2009], the number of the filter taps of CTF is set to $1/8$ of the T60. So given a T60 of $0.5$ s, the number of filter taps is roughly $0.5/8/0.016=3.9$, where $0.016$ is the STFT hop size in seconds. In other words, this setup echos our claims that $E$ is small.
>
> We will improve the sentence and cite the referred paper.
>
> > ll. 128 Typo in “comptue”
>
> Will correct!
>
> > Section 4.2 Mixture consistency has already been proposed in [Wisdom, 2019]. This paper should be clearly mentioned.
> Also, consider renaming to multichannel mixture consistency, since you’re using P channels and RTFs.
> > [Wisdom, 2019] Wisdom et al., Differentiable Consistency Constraints for Improved Deep Speech Enhancement, 2019.
>
> We now realize that it is a bad idea to use the same name, and we will change the name of our loss to "mixture-constraint" loss to differentiate it from the "mixture consistency" term proposed in [Wisdom, 2019].
>
> We will highlight their differences in the paper.
> As you mentioned, our loss differs in the number of channels used and, in addition, we filter the DNN estimates before loss computation.
> Another major difference is that [Wisdom, 2019] constrains the estimated sources to strictly add up to the mixture (see their Eq. (7) and (9)), while our method only ``encourages'' the filtered source estimate to add up to the mixture.
>
> > Section 4.3 Formulation is (6) is basically multichannel linear prediction with using estimated sources Z instead of the original mixtures Y. Relevant works solving a problem analogous to (6) should be at least briefly mentioned, such as earlier work in [Yoshioka, 2010] and many later works.
>
> > [Yoshioka, 2010] Yoshioka et al., Blind Separation and Dereverb. of Speech Mixtures by Joint Optimization, TASLP 2011.
>
> Will mention!
>
> Although (6) appears similar to conventional multichannel linear prediction (MCLP), we would like to emphasize that it has very different physical meanings. We consider that (6) does "forward filtering", where source estimates are filtered to approximate mixtures, while MCLP does "inverse filtering", where mixtures are filtered to approximate sources.
> This modification leads to non-trivial changes of the physical meanings of the computed filters (see also discussions in Section V.C of [A1] listed below).
>
> [A1] Z.-Q. Wang et al., Convolutive Prediction for Monaural Speech Dereverb. and Noisy-Reverb. Speaker Separation, in TASLP, 2021.
>
> > The main limitation is that the model does not take into account background or discrete noise, and is only demonstrated to work in presence of uncorrelated (measurement) noise. This severely limits applicability to using real-world recordings to train a separation model. It would be very useful to include results on robustness in presence of higher levels of noise. However, this is not essential and it may be out of scope of this paper.
>
> See our earlier response to this point. We also think this may be out of scope of this paper.
>
> > Another limitation for real-world use is the assumption of a static scenario (fixed relative IRs), which would not hold in real recordings.
>
> This is indeed a tricky issue, which is a common problem also existed in many other algorithms. We could model time-varying filters in some way, and we will investigate this in a future work.

---

### Author Rebuttal · Authors · 2023-08-09

We would like to thank the reviewers for their valuable feedback towards improving this manuscript.
Here we collectively address some common concerns raised by the reviewers:

$\textbf{1. Reasons for not comparing with MixIT in the first submission}$

We carefully considered using MixIT as a baseline, since MixIT also deals with unsupervised separation, but we realize that MixIT may not be a good baseline to UNSSOR out of the following considerations:

$\textbf{1.1.}$ MixIT needs to be trained on synthetic mixtures of mixtures (MoM), while UNSSOR is designed to be trained directly on existing mixtures. The two models would be trained on different training examples, and our concern is that this would make the comparison difficult. We hence consider methods that can be trained (or performed) directly on existing mixtures (such as IVA, spatial clustering and iRAS) as baselines.

$\textbf{1.2.}$ One could argue that we could, for example, mix the existing 2-speaker mixtures in SMS-WSJ to train MixIT, and compare it with UNSSOR trained directly on the existing 2-speaker mixtures in SMS-WSJ. However, this would require the mixtures used for creating each MoM to be recorded in the same room, by the same array, and at the same location in the room, each of which would incur restrictions to the dataset that can be used for training MixIT models, while UNSSOR does not have such restrictions.

$\textbf{1.3.}$ We could simulate a particular scenario, where, in each simulated room, we generate 4 speaker sources so that we can have several 2-speaker mixtures to create MoM for training MixIT models (for 4-speaker separation) and then compare the performance with the performance of UNSSOR trained on the same 2-speaker mixtures for 2-speaker separation. However, this simulated scenario would be very ideal for MixIT. Note that the eventual MixIT system should synthesize MoM based on real-recorded mixtures. Many times, the procedure for synthesizing MoM is very tricky, since real-recorded mixtures are usually not recorded in the same room, using the same array, and at the same location in the room.

$\textbf{1.4.}$ A possibly good way to compare UNSSOR and MixIT is by using real-recorded datasets such as AMI, AliMeeting and CHiME-{5,6,7} recorded in meeting scenarios where concurrent speech naturally happens. However, for both algorithms, to achieve good performance we need to solve many other problems (such as sparse speaker overlap, varying number of speaker, unknown number of speakers etc.) and there do not exist mature solutions yet; and, in addition, including the solutions to many other problems in this paper would make the paper much less focused. We hence lean towards leaving this investigation to a future study, and focus on showing the potential of UNSSOR, which avoids using synthetic MoM.

We will add these discussions and considerations to the paper.

$\textbf{2. Performance comparison with MixIT}$

Following the reviewers' suggestions, in this rebuttal, we have been trying to provide a performance comparison with MixIT.
However, due to time limit, the training cannot finish before the rebuttal deadline, and we will update the results during the discussion phase (by the end of Aug. 13 ET).

What we have been doing is to create the particular scenario (**ideal for MixIT**) described in paragraph $\textbf{1.3}$ above, where, for each existing SMS-WSJ mixture ($y = s_1 + s_2 + n$ where $n$ denotes noise), we randomly add two extra speakers in the same simulated room and use the same array placed at the same location so that we can have two 2-speaker mixtures (i.e., mixture 1: $s_1 + s_2 + n/2$ and mixture 2: $s_3 + s_4 + n/2$) to create MoM for training MixIT for 4-speaker separation.
The DNN architecture and training configurations for MixIT are the same as that in UNSSOR and PIT, the loss function is defined similarly on the real, imaginary and magnitude of the reconstructed mixtures, and the DNN can take single- or multi-channel input.
At run time, the trained MixIT model is used to separate the existing two-speaker mixtures in SMS-WSJ to four outputs, and we select the two outputs with the highest energy for evaluation. This way, the numbers can be pretty much directly compared with the existing ones obtained by UNSSOR.

We will update the results during the discussion phase (by the end of Aug. 13 ET). We will also add the results to the paper, if the paper is accepted.

---

> ### Author Response · Authors · 2023-08-19
>
> We would like to thank the reviewers for waiting for us to get the results of MixIT. We are excited to update that we have got the numbers.
>
> We create the particular scenario earlier described in the paragraph **1.3**.
> In detail, for each existing SMS-WSJ mixture ($y = s_1 + s_2 + n_1$, where $n_1$ denotes noise), we randomly place two more speakers in the same simulated room and use the same array placed at the same location so that we can have two 2-speaker mixtures (i.e., mixture 1: $s_1 + s_2 + n_1$ and mixture 2: $s_3 + s_4 + n_2$) to create MoM for training MixIT for 4-speaker separation.
> Similarly to $n_1$, $n_2$ is sampled such that the SNR between $s_3+s_4$ and $n_2$ is in the range of $[20, 30]$ dB.
>
> Similarly to Eq. (5) in the paper, the loss function for MixIT is defined on the real, imaginary and magnitude of each 2-speaker mixture weighted by the summation of the magnitude of the 2-speaker mixture.
>
> Similarly to our proposed technique, we can feed $1$-, $3$- or $6$-channel mixtures to TF-GridNet-based MixIT.
>
> At run time, the trained TF-GridNet-based MixIT model is used to separate the existing two-speaker mixtures in SMS-WSJ to 4 outputs, and we select the 2 outputs with the highest energy for evaluation.
> This way, the evalation results can be pretty much directly compared with the existing ones obtained by UNSSOR.
>
> The results on SMS-WSJ are shown below. They are not as good as the ones reported in Table 1-4 of the paper. For unknown reasons, 3ch MixIT fits the loss well but fails at separating speakers.
>
> |          | SDR (val.)| SDR | SI-SDR | PESQ | eSTOI
> |:-----: | :-----: | :----: | :----: | :----: | :----: |
> | 1ch MixIT | 6.6 | 6.6 | 6.3 | 2.21 | 0.691
> | 3ch MixIT | 0.1 | 0.1 | 0.0 | 1.91 | 0.608
> | 6ch MixIT | 8.2 | 8.0 | 7.8 | 2.43 | 0.744
>
> We will add the discussion and results to the paper.

---

### Decision · Program_Chairs · 2023-09-21

**Decision:**

Accept (poster)

**Comment:**

This paper proposed an algorithm named unsupervised neural speech separation by leveraging over-determined training mixtures.  It received acceptance from all reviewers, with reviewer ux7k particularly appreciating the rebuttal and changing his/her score from 5 to 6. There is no doubt that this paper is recommended for acceptance.